# Exploiting macrophage autophagy-lysosomal biogenesis as a therapy for atherosclerosis

Ismail Sergin[1], Trent D. Evans[1], Xiangyu Zhang[1], Somashubhra Bhattacharya[1], Carl J. Stokes[1], Eric Song[1], Sahl Ali[1], Babak Dehestani[1], Karyn B. Holloway[1], Paul S. Micevych[1], Ali Javaheri[1], Jan R. Crowley[2], Andrea Ballabio[3], Joel D. Schilling[1,4], Slava Epelman[5], Conrad C. Weihl[6], Abhinav Diwan[1], Daping Fan[7], Mohamed A. Zayed[8] & Babak Razani[1,4]

Macrophages specialize in removing lipids and debris present in the atherosclerotic plaque. However, plaque progression renders macrophages unable to degrade exogenous atherogenic material and endogenous cargo including dysfunctional proteins and organelles. Here we show that a decline in the autophagy–lysosome system contributes to this as evidenced by a derangement in key autophagy markers in both mouse and human atherosclerotic plaques. By augmenting macrophage TFEB, the master transcriptional regulator of autophagy–lysosomal biogenesis, we can reverse the autophagy dysfunction of plaques, enhance aggrephagy of p62-enriched protein aggregates and blunt macrophage apoptosis and pro-inflammatory IL-1β levels, leading to reduced atherosclerosis. In order to harness this degradative response therapeutically, we also describe a natural sugar called trehalose as an inducer of macrophage autophagy–lysosomal biogenesis and show trehalose's ability to recapitulate the atheroprotective properties of macrophage TFEB overexpression. Our data support this practical method of enhancing the degradative capacity of macrophages as a therapy for atherosclerotic vascular disease.

[1] Department of Medicine, Cardiovascular Division, Washington University School of Medicine, Campus Box 8086, 660 S Euclid Avenue, St Louis, Missouri 63110, USA. [2] Department of Medicine, Division of Endocrinology, Metabolism, and Lipid Research, Washington University School of Medicine, 660S Euclid Avenue, St Louis, Missouri 63110, USA. [3] Telethon Institute of Genetics and Medicine, Via Campi Flegrei 34, 80078 Pozzuoli, Naples, Italy. [4] Department of Pathology and Immunology, Washington University School of Medicine, Campus Box 8086, 660S. Euclid Avenue, St Louis, Missouri 63110, USA. [5] Peter Munk Cardiac Center, University Health Network, Toronto, Ontario, Canada M5G 2C4. [6] Department of Neurology, Washington University School of Medicine, Campus Box 8111, 660S. Euclid Avenue, St Louis, Missouri 63110, USA. [7] Department of Cell Biology and Anatomy, University of South Carolina School of Medicine, Columbia, South Carolina 29209, USA. [8] Department of Surgery, Washington University School of Medicine, 660S. Euclid Avenue, St Louis, Missouri 63110, USA. Correspondence and requests for materials should be addressed to B.R. (email: brazani@im.wustl.edu).

The macrophage is a central player in atherosclerotic progression. Insidious macrophage infiltration of the nascent plaque, phagocytosis of deposited lipid and cellular debris, foam cell formation, and progressive macrophage dysfunction and inflammatory signalling constitute some of the major events during atherogenesis. Understanding the cellular processes that underlie macrophage dysfunction remains an important area of investigation both scientifically and clinically, serving as the basis for future therapeutics. An area that has garnered substantial recent focus is autophagy, a highly evolutionarily conserved process with critical roles in the degradation and recycling of long-lived/damaged intracellular material including accumulated lipids[1,2]. We and others have shown that disruption of macrophage autophagy in mice (by deletion of the essential autophagy protein ATG5) leads to marked increases in atherosclerosis[3–5]. Proposed mechanisms for this observation include reductions in lipophagy and the delivery of cholesteryl esters for lysosome-mediated cholesterol efflux, hyperactivation of the inflammasome and IL-1β signalling, and increased cell death from accumulation of cytotoxic protein aggregates.

However, details regarding the autophagy dysfunction that occurs in atherosclerotic macrophages, the triggers that result in such dysfunction and whether stimulation of macrophage autophagy can be atheroprotective remain unclear. Using the accumulation of the autophagy marker p62/SQSTM1, a chaperone for selective autophagy of cargo such as protein aggregates, we have previously demonstrated that macrophages develop a progressive autophagy dysfunction in the developing plaque[3,6]. Regarding the trigger for autophagy dysfunction, a critical factor appears to be the functional status of macrophage lysosomes. Since lysosomes mediate the overall degradative capacity of cells including autophagosome processing, the development of lipid-induced lysosomal dysfunction in plaque macrophages is an important contributor to the observed autophagy deficiency[7–12]. Thus, attempts at reducing atherosclerosis by correcting the degradative capacity of macrophages would most likely succeed by stimulating the autophagy–lysosome system as a whole.

Recent discovery of the transcription factor TFEB as the predominant transcriptional activator of a broad network of autophagy and lysosomal genes (that is, master regulator of autophagy–lysosomal biogenesis) has enabled serious consideration of this idea[13,14]. A member of the MiT/TFE helix–loop–helix subfamily, TFEB, initiates an autophagy–lysosomal biogenesis programme, thus stimulating the overall degradative capacity of cells[14,15]. TFEB has also recently been demonstrated to increase lysosomal lipid catabolism, lipolysis and cellular fatty-acid oxidation[16,17]. This provides an exciting new way to reverse macrophage dysfunction by enhancing autophagy–lysosomal function. By overexpressing TFEB in cultured macrophages, we recently showed the ability of autophagy–lysosomal biogenesis to reverse the autophagy–lysosomal dysfunction instigated by atherogenic lipids and to have several functional benefits such as induction of cholesterol efflux, dampening of inflammasome activation and clearance of polyubiquitinated protein aggregates[12]. It remains unclear how autophagy–lysosome dysfunction manifests and progresses in atherosclerotic plaques, and whether enhancing macrophage autophagy–lysosomal biogenesis can be effective against atherosclerosis.

A second crucial question is whether the macrophage autophagy–lysosomal degradation system can be stimulated via pharmacological means to confer atheroprotection. Trehalose is a non-reducing natural disaccharide (α,α-1,1-glucoside) synthesized endogenously by non-mammalian organisms such as insects, crustaceans and certain plants[18]. Present in high concentration in these organisms, trehalose is thought to provide protection against environmental stresses such as osmotic and temperature shocks by stabilizing biomolecules[18,19]. In fact, the pharmaceutical industry uses trehalose as a stabilizer excipient in numerous medicines. In the nutraceutical industry, trehalose has even been used as a sweetener because of its mild sweetness as compared to its other closely related non-reducing disaccharide sucrose[20,21]. Interestingly, there is a sizable body of literature demonstrating trehalose's ability to induce autophagy, although the mechanism remains unknown[22–29]. This property has been exploited by several studies in the neurodegeneration field to reduce aggregate formation in mouse models of Huntington's, Parkinson's and Lou Gehrig's (amyotrophic lateral sclerosis) disease[22,24,26]. Owing to trehalose's ability to induce autophagy and ameliorate various protein aggregation neurodegenerative diseases and our finding of TFEB as an autophagy inducer particularly of protein aggregates, we became interested in evaluating the effects of trehalose in the induction of macrophage TFEB, autophagy–lysosomal biogenesis and atherosclerosis.

In the first part of this manuscript, we characterize the progressive macrophage autophagy–lysosomal dysfunction that occurs in both murine and human atherosclerotic plaques and mechanistically explore how macrophage-specific TFEB overexpression can rescue these defects to reduce atherosclerotic plaques. In the second part of this manuscript, we leverage the prodegradative response of macrophages for therapeutic purposes by describing trehalose as a viable and practical method of stimulating autophagy–lysosomal biogenesis to reduce atherosclerosis.

## Results

**Atherosclerotic plaques develop autophagy dysfunction.** Although the link between deficient macrophage autophagy and increased atherosclerosis has been reported by several groups, including ours[3–6], the exact nature of the autophagy dysfunction that takes place in atherosclerotic macrophages is unclear. Macrophages of the atherosclerotic plaque are known to be the predominant expressors of two commonly used autophagy markers, LC3 (an autophagosome coat protein and a direct marker of autophagy progression) and p62 (a selective autophagy chaperone that detects and delivers large biomolecules such as protein aggregates or organelles to autophagosomes)[3]. However, details about the expression of these markers with relation to plaque progression and their correlation with the observed autophagy dysfunction remain unknown. Since LC3 levels correlate with the progression of autophagy and the accumulation of p62 levels correlate with stalled or dysfunctional autophagy[30], we first conducted detailed analysis of immunostained atherosclerotic plaques from atheroprone ApoE-null mice and atherosclerotic human carotid samples.

First, we compared the levels and co-localization of LC3 and p62 in aortic root plaques of ApoE-null mice at two stages of atherogenesis ('early' versus 'advanced' lesions as defined by the duration of western diet feeding). LC3 levels were significantly lower in the more advanced lesions, whereas p62 intensity remained elevated in both early and advanced lesions (Fig. 1a–c). In early lesions, LC3 levels were still high, and much of the p62 showed co-localization with LC3; this co-localization markedly waned in advanced lesions (Fig. 1d,e). Consistent with our previously reported observation, p62 (a chaperone for polyubiquitinated protein aggregates) showed high co-localization with polyubiquitin stains in both lesion types (Supplementary Fig. 1a)[6]. Overall, these data indicate that the observed dysfunction in plaque autophagy is manifested as a constellation of events (that is, p62 accumulation, decline in LC3/autophagosome formation and an inability to co-express autophagy markers).

We were interested in determining whether our findings in mouse models are recapitulated in human atherosclerosis and repeated a similar analysis in human atherosclerotic plaques obtained from discarded carotid endarterectomy (CEA) specimens. CEA lesions were dissected immediately postoperatively into regions devoid of disease or regions with minimal or maximal disease, for LC3 and p62 staining. In areas lacking atherosclerosis, we only detected LC3 without any evidence for p62 or ubiquitin accumulation by immunofluorescence (IF), consistent with the presence of basal of autophagy without aberrant accumulation of autophagic cargo (Supplementary Fig. 1b). The accumulation of p62 was detected in minimally

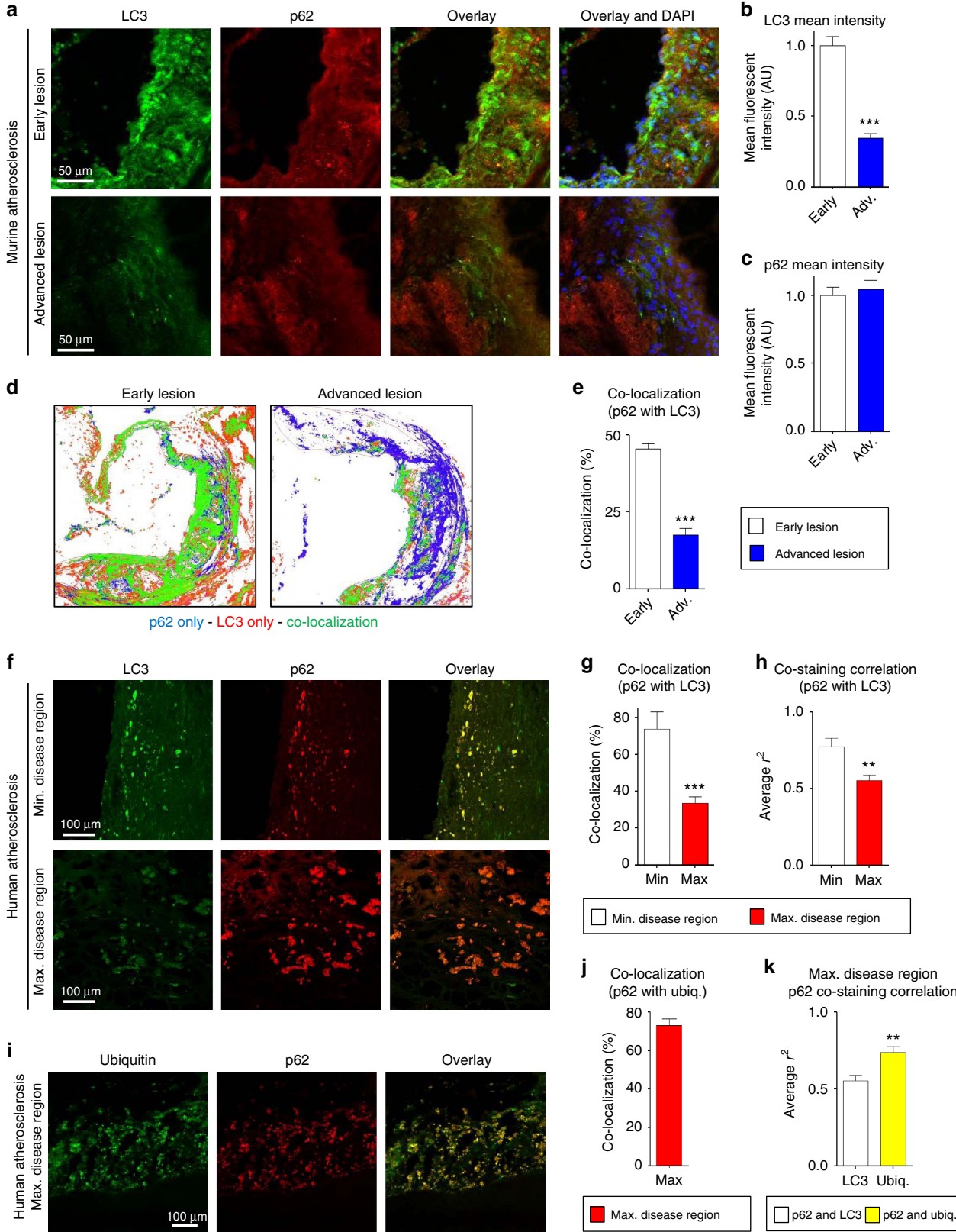

diseased regions and increased (that is, p62 staining increased in both size and number) in maximally diseased regions, although it did not reach statistical significance (Fig. 1f and Supplementary Fig. 1c). In contrast, the intensity of LC3 staining was both reduced (Fig. 1f and Supplementary Fig. 1d) and developed poor localization with p62 (Fig. 1g,h) in maximally diseased regions. Given the lower co-localization of p62 and LC3, we also evaluated the staining pattern of p62 and polyubiquitin in the same samples and found the large majority of p62 co-localized with polyubiquitin (Fig. 1i,j). Technical limitations precluded co-staining of plaques with LC3, p62 and polyubiquitin in unison and direct correlation between all markers. However, comparison of co-staining correlations between p62 and LC3 and p62 and polyubiquitin demonstrated significantly better correlation for p62 and polyubiquitin (Fig. 1k). Taken together, these data suggest that in advancing atherosclerotic plaques the formation and stagnation of p62-enriched polyubiquitinated aggregates is favoured over clearance and is a hallmark feature of the autophagy dysfunction in atherosclerosis.

**Macrophage TFEB overexpression rescues plaque autophagy.** The progressive decrease in LC3 levels and p62/LC3 co-localization as well as the accumulation of polyubiquitinated proteins with advancing plaque formation led us to investigate methods by which the autophagic degradation machinery can be stimulated. TFEB is the predominant transcription factor capable of inducing coordinated expression of autophagy–lysosomal genes and the prodegradative response in several cell types including macrophages[12–14]. We utilized a previously described tissue-specific overexpression model of TFEB in mice and created macrophage-specific TFEB-transgenic mice by conducting crosses with mice expressing Cre under the control of Lysozyme-M promoter (hereafter referred to as mϕTFEB-TG)[12,13]. Thioglycollate-elicited peritoneal macrophages (hereafter referred to as macrophages) from these mice showed increased TFEB expression, more TFEB nuclear localization and TFEB-induced target gene expression including p62 and LC3 (Fig. 2a–c, Supplementary Figs 2a and 12a). We also found enhanced autophagic flux in TFEB-TG (tissue-specific transgenic expression of TFEB) macrophages by observing baseline elevations in LC3-I levels with enhanced conversion to LC3-II upon bafilomycin incubation (a lysosomal inhibitor/blocker of autophagosome degradation; Fig. 2d and Supplementary Fig. 11a). In addition, p62 levels were persistently higher in agreement with our gene expression results (Fig. 2c,d). We crossed mϕTFEB-TG mice with green fluorescent protein-tagged LC3 (GFP-LC3)-expressing mice to be able to further monitor autophagy in macrophages by live imaging. TFEB-TG macrophages showed significantly more GFP-LC3-positive area in 20 min of live imaging, suggesting induced autophagosome formation even under unstimulated conditions (Fig. 2e, Supplementary Fig. 2b,c and Supplementary Movies 1 and 2). When these macrophages were incubated with bafilomycin, even higher GFP-LC3 fluorescence was observed consistent with TFEB-induced increases in autophagic flux (Fig. 2f, Supplementary Fig. 2d,e and Supplementary Movies 3 and 4). Increased flux was also corroborated by GFP-LC3 dots showing more co-localization with the lysosome marker Lysotracker in TFEB-TG macrophages (Fig. 2g, Supplementary Fig. 2f–h and Supplementary Movies 5 and 6). In agreement with our real-time live imaging experiments, we observed a significantly higher autophagic flux in TFEB-TG macrophages stained by LC3 immunocytochemistry at both baseline and after 3 h of treatment with another blocker of lysosomal degradation chloroquine (Fig. 2h). In keeping with TFEB's ability to induce the autophagy–lysosome system as a whole, we also detected increases in LAMP1 expression (a commonly used marker of lysosomes and lysosomal mass) both in cultured peritoneal macrophages and splenic macrophages derived from mϕTFEB-TG mice (Fig. 2i). In conclusion, these data suggest that overexpressing TFEB is a sufficient method to induce the autophagosome formation and autophagy–lysosomal biogenesis in macrophages.

We next asked whether macrophage-specific TFEB overexpression can induce autophagy and autophagy–lysosomal biogenesis in atherosclerotic plaques in vivo. We crossed mϕTFEB-TG mice with pro-atherogenic ApoE-KO mice and initiated plaque formation by western diet feeding. TFEB expression as gauged by IF staining of aortic roots was significantly elevated and coincided with plaque macrophages (Fig. 3a,b). More importantly, TFEB nuclear localization was particularly elevated in the atherosclerotic plaques of mϕTFEB-TG mice using two independent TFEB-specific antibodies (Fig. 3c and Supplementary Fig. 3a). The expression of both LC3 and p62 was also increased in mϕTFEB-TG aortic roots in agreement with our in vitro macrophage data (Fig. 3d,e). Interestingly, in direct contrast to what we observed in the progressive atherosclerosis of mice models and the maximally diseased regions of human plaques (Fig. 1), mϕTFEB-TG atherosclerotic plaques showed remarkably higher p62-LC3 co-localization and co-staining correlation than control atherosclerotic lesions (Fig. 3f,g). Macrophages from mϕTFEB-TG atherosclerotic aortas analysed by fluorescence activated–cell sorting (FACS) analysis also displayed enhanced co-expression of autophagy and lysosomal markers as gauged by co-staining of LC3 and p62 as well as LC3 and Lamp2 (Fig. 3h,i). Overall, these data suggest that macrophage-specific TFEB overexpression is a viable approach to induce autophagy and autophagy–lysosomal biogenesis, reprogramme plaque macrophages to co-associate autophagy markers and their cargo, and reverse the autophagy dysfunction observed in advancing atherosclerosis.

**Macrophage TFEB overexpression reduces atherosclerosis.** In order to assess the effect of macrophage TFEB overexpression on plaque progression, we fed cohorts of control and mϕTFEB-TG (on a pro-atherogenic ApoE-null background) a

**Figure 1 | Mouse and human atherosclerotic plaques develop features of a progressive autophagy dysfunction.** (**a–c**) Representative immunofluorescence images of early-stage and more advanced atherosclerotic (ApoE-KO) aortic roots co-stained with antibodies against LC3 and p62. Early and advanced lesions were obtained from ApoE-KO mice fed a western diet for <2 months and 3–4 months, respectively (scale bar, 50 μm (**a**)). The mean intensity for LC3 and p62 stainings were analysed ($n = 5$ mice for each group; **b,c**). (**d,e**) Co-localization of LC3 and p62 was also analysed in the same aortic roots. Representative co-localization images are shown from early and more advanced lesions (green indicates LC3/p62 co-localized, red indicates LC3-positive, and blue indicates p62-positive areas) (**d**). LC3/p62 co-localization is quantified as per cent of total signal (**e**). (**f–k**) Immunofluorescence analysis of human carotid endarterectomy specimens ($n = 8$), which are separated as maximally- and adjacent minimally diseased regions (scale bar, 100 μm). Specimens were co-stained with LC3 and p62 (**f**), and co-localization (**g**) as well as correlation of staining intensity (**h**) between maximally and minimally diseased regions are quantified. Maximally diseased human atherosclerotic regions were co-stained for p62 and polyubiquitinated proteins (FK-1 antibody; **i**), and co-localization quantified (**j**). (**k**) Graph represents a comparison of the staining correlation between p62/LC3 versus p62/ubiquitin(FK-1) in maximally atherosclerotic regions. For all graphs, data are presented as mean ± s.e.m. **$P < 0.01$, ***$P < 0.001$, two-tailed unpaired t-test. Max, maximum; Min, minimum; Ubiq., ubiquitination.

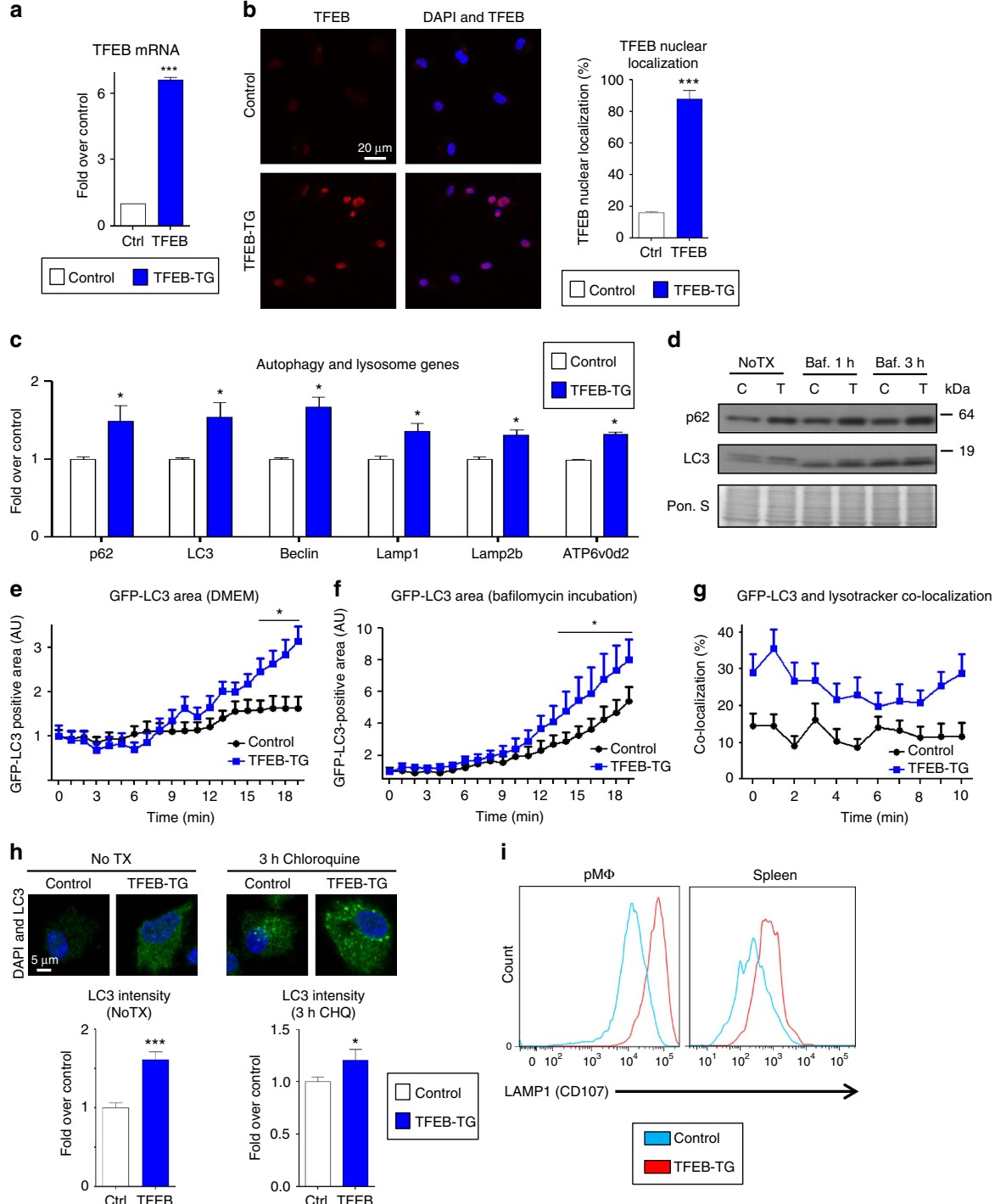

**Figure 2 | TFEB overexpression induces autophagy and autophagy–lysosomal biogenesis in macrophages.** (**a–c**) Control and TFEB-overexpressing (TFEB-TG) thioglycollate-elicited peritoneal macrophages (hereafter referred to as macrophages) were assessed as follows: transcript levels of (**a**) TFEB and (**c**) several autophagy–lysosome markers were evaluated by quantitative polymerase chain reaction (qPCR, $n \geq 3$ independent wells). (**b**) TFEB nuclear localization was assessed by immunofluorescence staining and quantified as percentage of TFEB-positive nuclei ($n \geq 40$ cells per group, scale bar, 20 μm). (**d**) Western blot analysis of p62 and LC3 in TFEB-TG macrophages after bafilomycin (200 nM) treatment for indicated times (C, control; T, TFEB-TG). (**e–g**) Control and TFEB-TG macrophages also co-expressing GFP-LC3 were evaluated by live imaging (every 30 s for the indicated times) while being incubated with either (**e**) DMEM, (**f**) bafilomycin (200 nM) or after staining with (**g**) Lysotracker-red. Graphs represent (**e,f**) GFP-LC3-positive areas or (**g**) per cent of GFP-LC3 co-localized with Lysotracker-red over the indicated times. For **e,f** each time point is compared with the control GFP-LC3 group ($n \geq 10$ cells for each treatment). (**h**) LC3 levels and the intracellular pattern were analysed by immunofluorescence staining of baseline (NoTX) or after 3 h of 10 μM chloroquine incubation. Graphs represent the mean LC3 intensity ($n = 16$–46 cells, scale bar, 5 μm). (**i**) FACS analysis of peritoneal and splenic macrophages from control or macrophage-specific TFEB-TG mice for LAMP1 expression. For all graphs, three independent experiments were performed; data presented as mean ± s.e.m. *$P < 0.05$, ***$P < 0.001$, two-tailed unpaired $t$-test. Baf,bafilomycin; CHQ, chloroquine; Ctrl, control; DAPI, 4,6-diamidino-2-phenylindole.

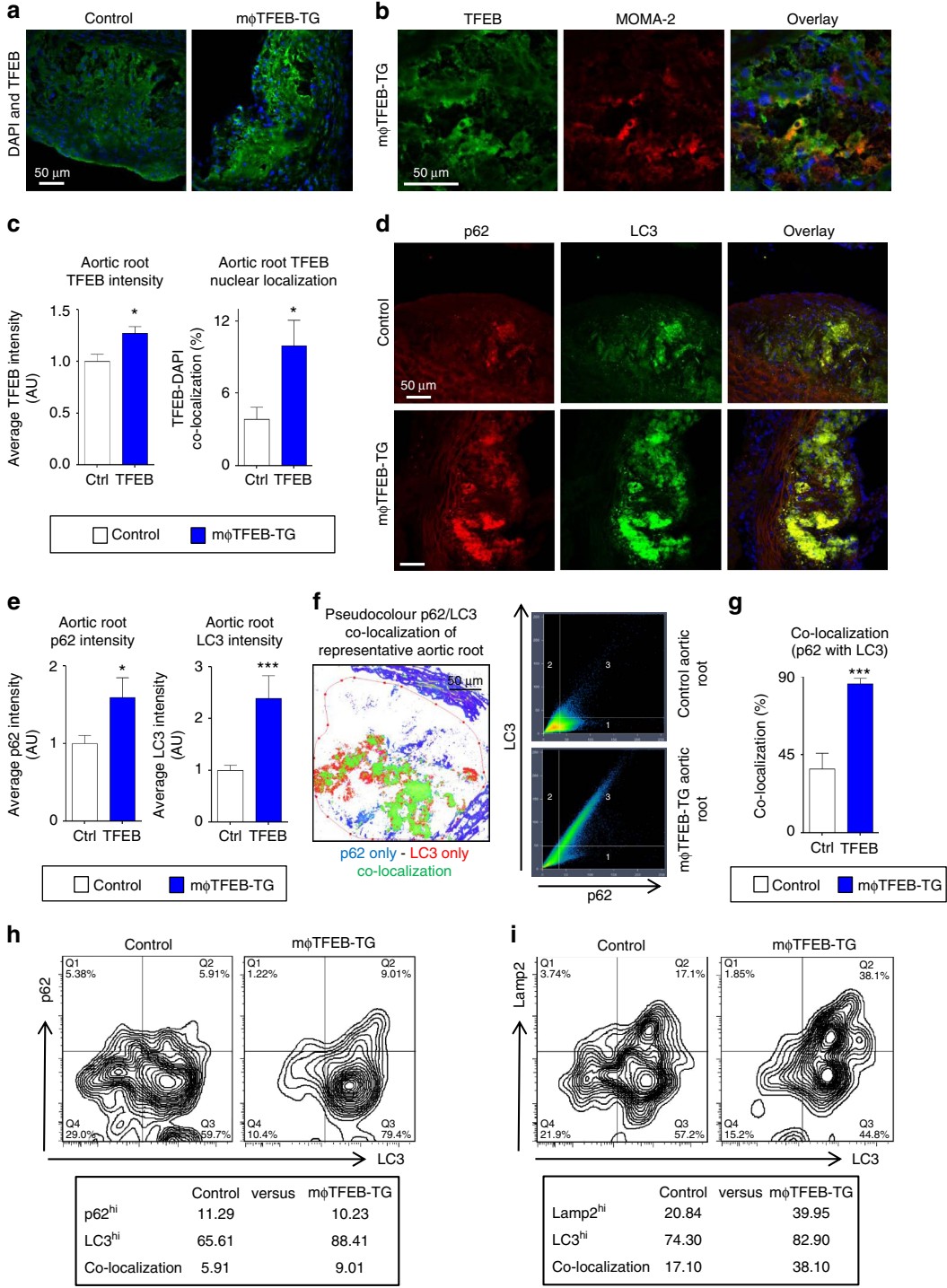

**Figure 3 | TFEB overexpression in macrophages induces the autophagy markers LC3 and p62 and restores their co-localization in atherosclerotic aortic roots.** (**a,b**) Representative immunofluorescence images of atherosclerotic aortic roots (2 months' western diet) from control and mϕTFEB-TG mice (ApoE-null background) stained with antibodies against TFEB (**a**), TFEB and MOMA-2 (**b**; scale bar, 50 μm). (**c**) Quantification of the average TFEB intensity and co-localization with nuclear marker DAPI (n = 4-5 mice per group). (**d**) Representative immunofluorescence images of atherosclerotic aortic roots from control and mϕTFEB-TG mice stained with p62 and LC3 (scale bar, 50 μm). (**e**) Quantification of the p62 and LC3 average intensity from control and mϕTFEB-TG-stained roots (n = 13–14 mice per group). (**f**) Representative pseudocolour image of these p62/LC3 images (green represents co-localization) and graph depicting the increased p62/LC3 correlation seen in a representative mϕTFEB-TG as compared to a control lesion (scale bar, 50 μm). (**g**) Quantification of the p62/LC3 co-localization from control and mϕTFEB-TG-stained roots shown (n = 13–14 mice per group). (**h,i**) FACS analysis of aortic macrophages isolated from atherosclerotic aortas of Control or mϕTFEB-TG mice (western diet-fed ApoE-KO background, n = 3–4 pooled aortas) and stained for either (**h**) p62 and LC3, or (**i**) Lamp2 and LC3 antibodies (per cent of macrophages expressing each marker is shown below plots). For all graphs, data are presented as mean ± s.e.m. *P < 0.05, ***P < 0.001, two-tailed unpaired t-test.

Western diet for 8 weeks and quantified atherosclerotic lesion formation and parameters of plaque complexity (Fig. 4a details this study). mφTFEB-TG mice did not show any difference on serum cholesterol or other common serum metabolites compared to controls (Fig. 4b and Supplementary Fig. 4a,b). Lesion quantitation revealed mφTFEB-TG mice were significantly

protected from atherosclerosis at both the level of the aortic root and whole aorta (Fig. 4c,d). Several features of plaque complexity were also concomitantly reduced: macrophage-positive and necrotic core areas were slightly reduced but the apoptotic-positive areas (as assessed by TUNEL staining) and the combined apoptotic/necrotic-positive areas were markedly

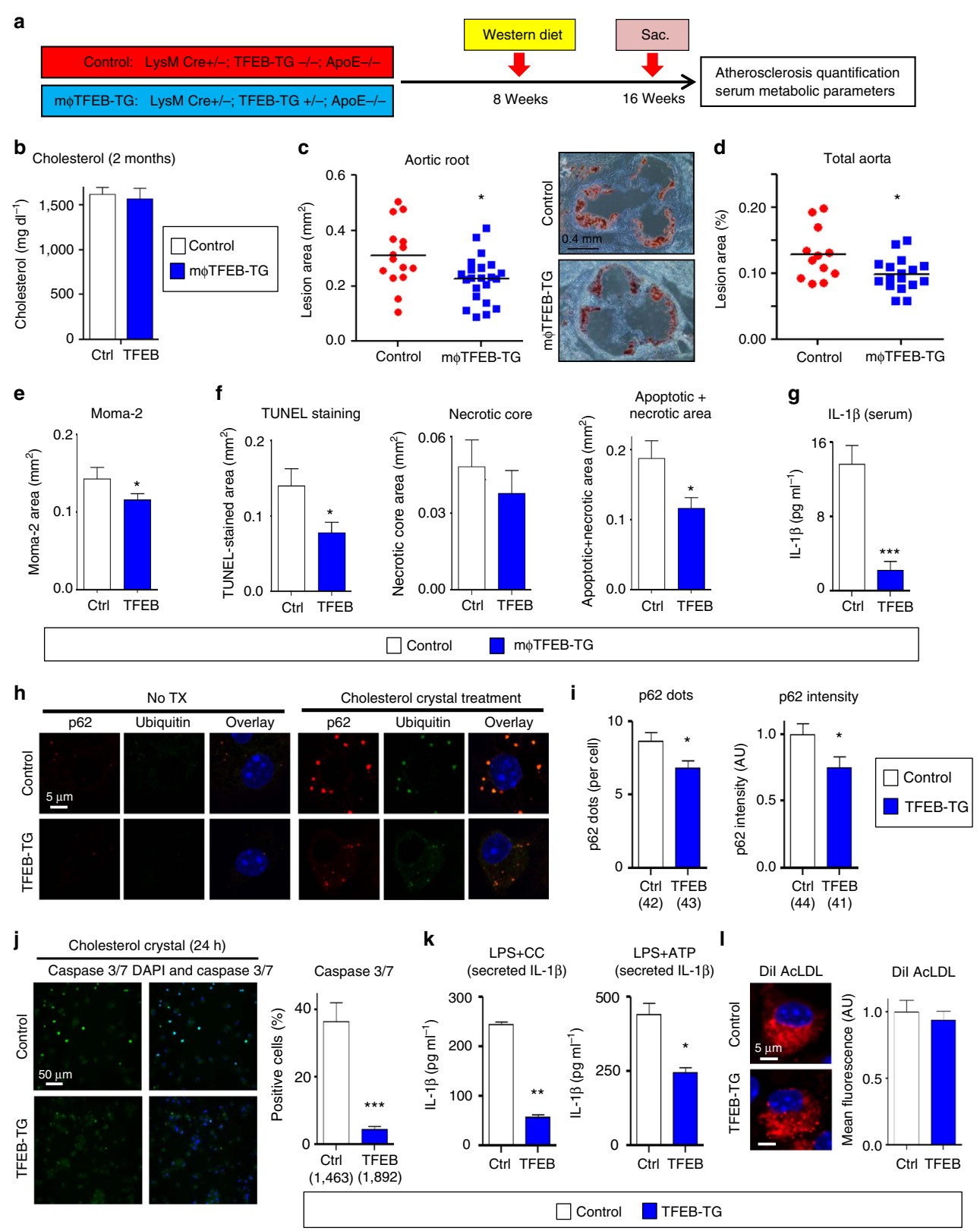

reduced contributing to an approximately 50% reduction in lesion area (Fig. 4e,f and Supplementary Fig. 4c–e). The pro-inflammatory cytokine interleukin (IL)-1β, which has an inverse correlation with macrophage autophagy, was also substantially reduced in the serum of mɸTFEB-TG mice (Fig. 4g). Given the ability of macrophage TFEB to induce autophagy and autophagy chaperones such as p62, we next evaluated the functional significance of some of these findings in cultured macrophages. In a recent detailed study of p62 in atherosclerosis, we have shown that atherogenic lipids such as cholesterol crystals render the macrophage autophagy–lysosome system dysfunctional and lead to accumulation of p62-enriched polyubiquitinated protein aggregates[6]. TFEB-TG macrophages reversed this effect and significantly reduced the accumulation of inclusion bodies (Fig. 4h,i). As we have previously shown, these inclusions are mostly composed of polyubiquitinated proteins, are largely devoid of significant amounts of lipids (Supplementary Fig. 4f) and disrupted autophagy of these p62-enriched aggregates is associated with increased apoptosis and inflammasome/IL-1β activation[6]. TFEB-TG macrophages also ameliorated these deleterious downstream effects by markedly reducing cholesterol crystal-induced macrophage apoptosis (Fig. 4j) and IL-1β secretion induced by various inflammasome-activating stimuli (Fig. 4k and Supplementary Fig. 4g). The anti-inflammatory phenotype of TFEB-TG macrophages appeared to be selective to IL-1β signalling because tumour necrosis factor secretion was not changed in those macrophages (Supplementary Fig. 4h). TFEB-TG macrophages also did not appear to show differences in lipid accumulation or foam cell formation upon challenge with modified lipids (Fig. 4l and Supplementary Fig. 4i–k) despite having a predicted rise in the activity of lysosomal acid lipase, a known transcriptional target of TFEB (Supplementary Fig. 4l)[12].

**TFEB-driven atheroprotection is autophagy and p62 dependent.** Our characterization of mɸTFEB-TG mice *in vivo* and cultured macrophages *in vitro* demonstrates TFEB's ability to increase macrophage autophagy and autophagy–lysosomal biogenesis, increase aggrephagy and the clearance of p62-enriched protein aggregates, decrease macrophage apoptosis and the pro-inflammatory cytokine IL-1β, and decrease atherosclerosis and plaque complexity. In order to determine whether the TFEB-autophagy-p62-mediated effects are causally linked to the observed reduction in atherosclerosis, we developed two new lines of mice by crossing mɸTFEB-TG mice with either macrophage-specific autophagy-deficient (mɸATG5-KO) or p62-deficient

(p62-KO) mice (on a pro-atherogenic ApoE-null background). Autophagy remained fully inactive in ATG5-KO macrophages even with concomitant TFEB overexpression as gauged by absence of the autophagosome marker LC3-II (Supplementary Figs 5a and 12b).

Consistent with TFEB's dependence on autophagy, the dual mɸTFEB-TG/ATG5-KO mice were no longer protected from atherosclerosis (Fig. 5a) with similar serum cholesterol and other common serum metabolites to controls (Supplementary Fig. 5b–d). We also observed no differences in parameters of plaque complexity (Fig. 5b,c). In addition, dual TFEB-TG/ATG5-KO macrophages subjected to atherogenic lipids were no longer able to (1) reduce the number of p62-enriched protein aggregates or (2) blunt the degree of apoptosis (Fig. 5d–f). We were surprised to find out, however, that TFEB retained its ability to specifically diminish the inflammasome/IL-1β levels even in the absence of autophagy, suggesting an independent mechanism that does not appear to be relevant to the mechanism of plaque reduction (Fig. 5g and Supplementary Fig. 5e,f).

Our findings with dual mɸTFEB-TG/ATG5-KO macrophages were mirrored with dual mɸTFEB-TG/p62-KO mice where again TFEB's protective effects on atherosclerosis and lesion complexity were abrogated in the absence of p62 (Fig. 6a–c and Supplementary Fig. 6a–c). As we have described before, the absence of p62 in macrophages leads to a disruption and further accumulation of polyubiquitinated proteins upon atherogenic lipid treatment[6]. TFEB overexpression was unable to reverse this effect, resulting in a similar degree of polyubiquitinated proteins in dual TFEB-TG/p62-KO macrophages (Fig. 6d,e). Consistent with the involvement of cell death pathways, TFEB overexpression was unable to prevent atherogenic lipid-induced apoptosis (Fig. 6f). With regard to the inflammasome and IL-1β, we were again surprised to discover that TFEB's ability to dampen IL-1β secretion was completely independent of p62 and reaffirmed the notion that TFEB mediates these IL-1β-suppressive effects via an alternative mechanism that appears to have no physiological significance for atherosclerotic plaque formation (Fig. 6g and Supplementary Fig. 6d,e).

**Trehalose induces macrophage autophagy–lysosomal biogenesis.** In order to leverage the therapeutic benefit of TFEB activation in macrophages, we were interested in evaluating compounds capable of stimulating a similar response. The natural disaccharide trehalose has been reported to have autophagy-inducing and protein aggregate-reducing effects that parallel those observed for TFEB, which led us to assess trehalose's ability

**Figure 4 | Macrophage-specific TFEB overexpression is atheroprotective.** (**a**) Experimental protocol and mouse cohorts used for assessment of atherosclerosis. (**b–g**) Control and mɸTFEB-TG mice (all on ApoE-KO background) were fed a western diet for 2 months for lesion development. (**b**) Serum cholesterol levels at 2 months ($n \geq 14$ mice per group). (**c**) Quantification of atherosclerotic plaque burden using Oil Red O-stained aortic root sections with representative roots shown on right (scale bar, 0.4 mm) and (**d**) *en face* analysis of whole aorta (statistical significance of differences calculated using Mann–Whitney $U$-test). (**e**) Macrophage content in aortic root sections was analysed by MOMA-2-positive area ($n \geq 12$ mice per group). (**f**) Apoptotic, necrotic core and combined apoptotic–necrotic core areas of aortic root sections were determined by quantifying TUNEL immunofluorescence staining, acellular areas or the combined area, respectively ($n \geq 12$ mice per group). (**g**) Serum IL-1β concentration was measured from $n \geq 6$ independent samples derived by pooling serum from two to three mice per sample ($> 12$ mice per group). (**h**) Control and TFEB-TG macrophages treated with or without cholesterol crystals for 24 h and stained using DAPI (nuclei) and antibodies against polyubiquitinated proteins (FK-1) and p62 (scale bar, 5 μm). (**i**) Quantification of average p62/ubiquitin-positive dots and average p62 intensity per cell from immunofluorescence experiment described in **h** (numbers of cells under each bar). (**j**) Control and TFEB-TG macrophages were incubated with cholesterol crystals and per cent of caspase 3/7-positive cells quantified in three independent experiments. Representative immunofluorescence images are shown on left and numbers of cells shown under each bar (scale bar, 50 μm). (**k**) Control and TFEB-TG macrophages were treated with LPS (lipopolysaccharide) + cholesterol crystals (hereafter referred to as LPS + CC) for 24 h or with LPS followed by ATP for 3 h. Culture media were assayed for IL-1β by ELISA ($n = 3$ independent wells for each treatment). (**l**) Control and TFEB-TG macrophages were treated with DiI-acetylated LDL for 12 h and intracellular lipid accumulation quantified by immunofluorescence microscopy ($n \geq 187$ cell per group, scale bar, 5 μm). For all graphs, data are presented as mean ± s.e.m. *$P < 0.05$, **$P < 0.01$, ***$P < 0.001$, two-tailed unpaired $t$-test, except **c**,**d**.

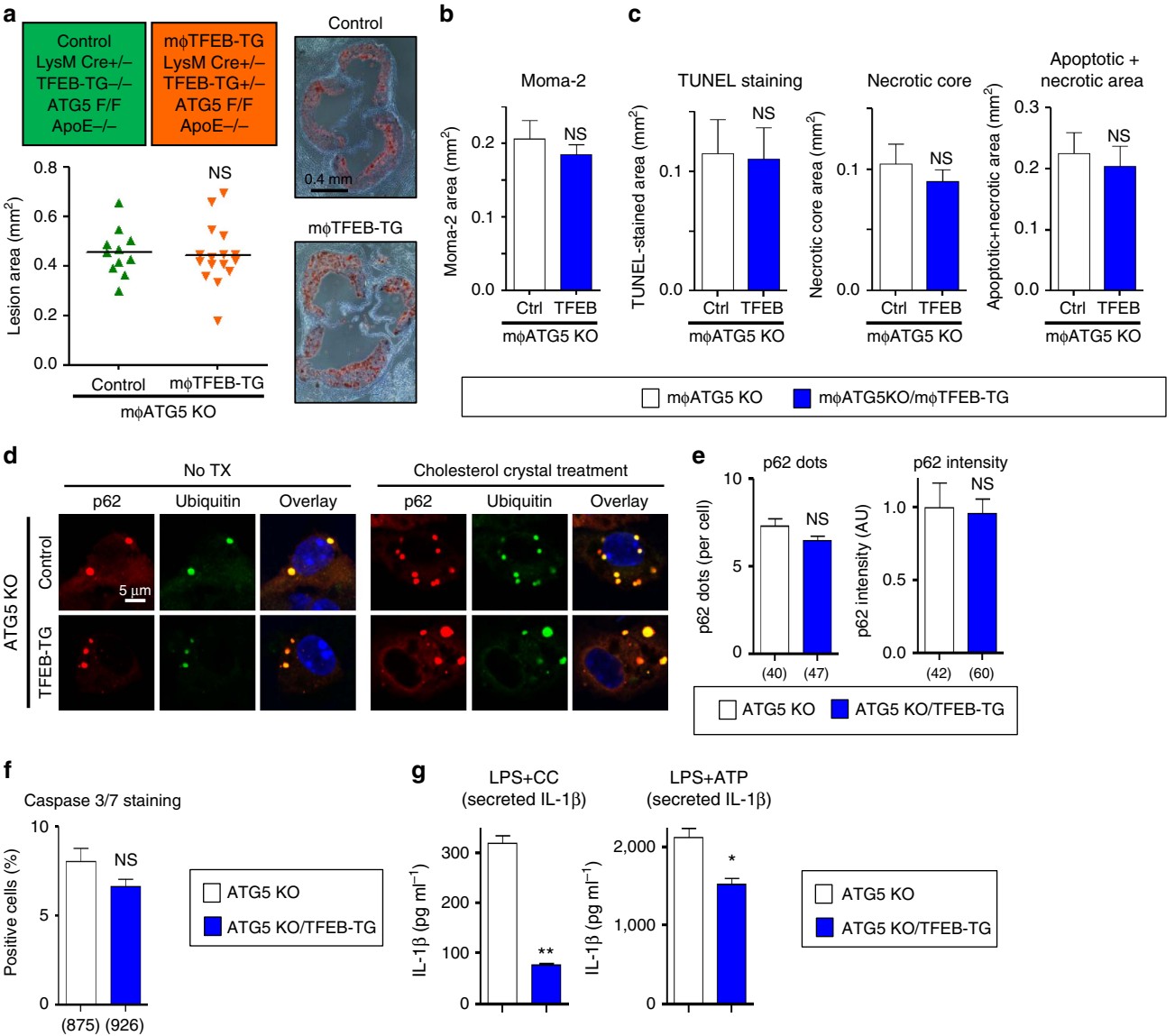

**Figure 5 | Macrophage-specific TFEB overexpression requires an intact autophagy pathway for atheroprotection including efficient clearance of polyubiquitinated protein aggregates and reductions in macrophage apoptosis.** (**a**) Cohorts of control and mφTFEB-TG mice (all on mφATG5-KO and ApoE-KO background) were fed a western diet for 2 months to develop lesions—exact genotypes are provided at the top of the graph. Atherosclerotic plaque burden was quantified by computer image analysis of Oil Red O-stained aortic root sections with representative Oil Red O-stained aortic roots shown on right (scale bar, 0.4 mm; statistical significance of differences was calculated using Mann–Whitney U-test). (**b**) Macrophage content in aortic root sections was analysed by immunofluorescence staining using an antibody against MOMA-2 ($n \geq 10$ mice per group). (**c**) Apoptotic, necrotic core and combined apoptotic–necrotic core areas of aortic root sections were determined by quantification of TUNEL immunofluorescence staining, acellular areas or a combination of the two, respectively ($n \geq 10$ mice per group). (**d**) Immunofluorescence images of ATG5-KO and ATG5-KO/TFEB-TG macrophages treated with or without cholesterol crystals for 24 h and stained using DAPI (nuclei) and antibodies against polyubiquitinated proteins (FK-1) and p62 (scale bar, 5 μm). (**e**) Quantification of average p62/ubiquitin-positive dots and average p62 intensity per cell from immunofluorescence experiment described in **d** (numbers of quantified cells are shown under each bar). (**f**) ATG5-KO and ATG5-KO/TFEB-TG macrophages were incubated with cholesterol crystals and the per cent of caspase 3/7-positive cells was quantified in three independent experiments (numbers of quantified cells are shown under each bar). (**g**) ATG5-KO and ATG5-KO/TFEB-TG macrophages were treated with LPS + CC for 24 h or with LPS, followed by ATP for 3 h. Culture media were assayed for IL-1β by ELISA ($n = 3$ independent wells for each treatment). For all graphs, data are presented as mean ± s.e.m. *$P < 0.05$, **$P < 0.01$, NS: not significant, two-tailed unpaired t-test, except **a**.

to stimulate macrophage autophagy, autophagy–lysosomal biogenesis and downstream effects on atherosclerosis.

We first aimed to determine a therapeutically relevant trehalose dose that could be tested both in macrophages *in vitro* and then in our atherosclerosis mouse models *in vivo*. This was an important initial evaluation due to the discrepancy in prior literature on what constitutes a physiologically effective dose of trehalose for autophagy induction; many studies use concentrations as high as 100 mM in cultured cells that appear to differ greatly from doses likely achievable *in vivo*[22–28]. We thus administered trehalose to a cohort of mice intraperitoneally (i.p. 3 g kg$^{-1}$ body weight), a dose similar to that used in prior studies involving mouse models[22,26,28,29], and determined simple pharmacokinetics of serum trehalose (Fig. 7a). Over a 2 h time

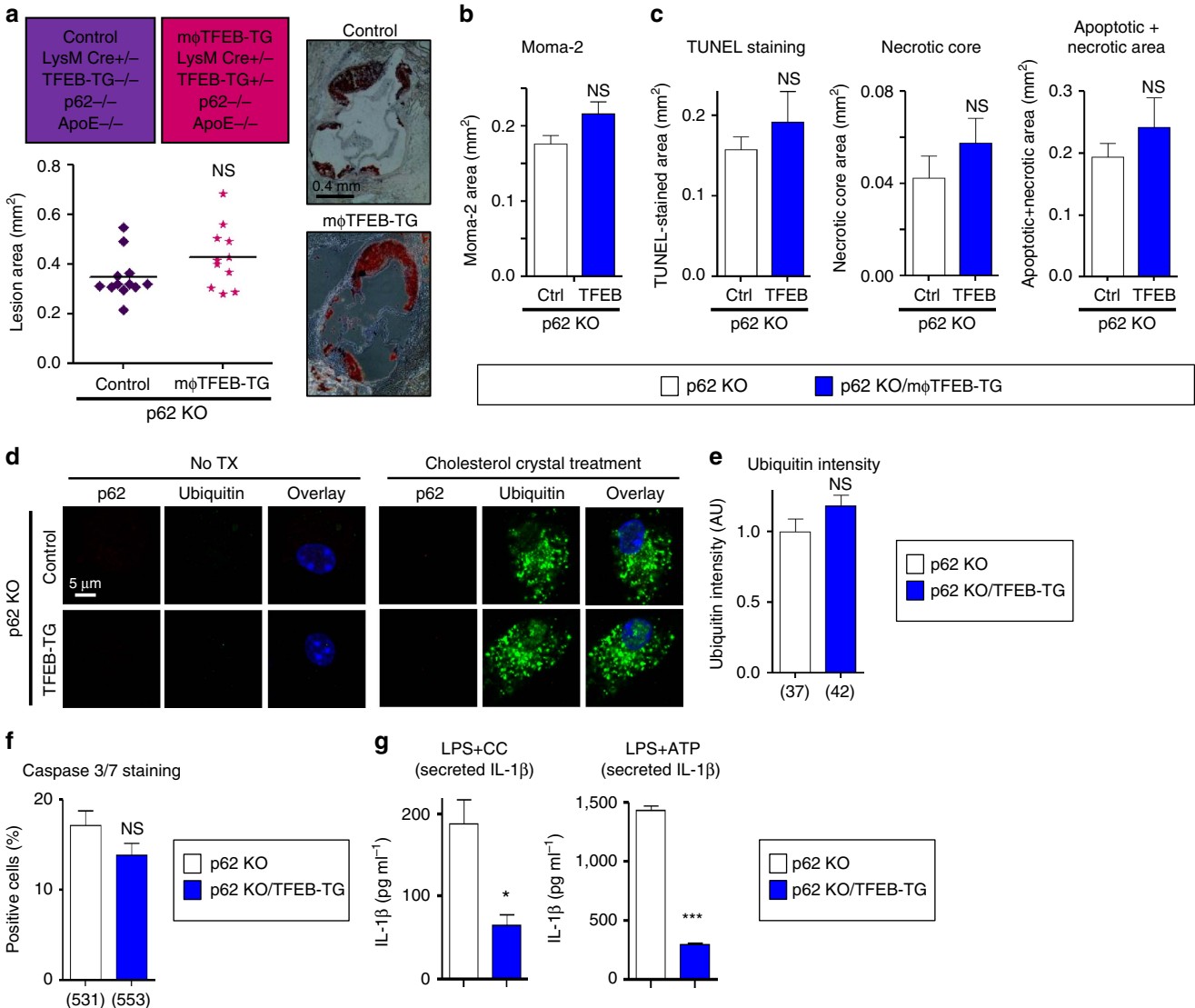

**Figure 6 | The atheroprotective effect of macrophage-specific TFEB overexpression is also p62-dependent. (a)** Cohorts of control and mφTFEB-TG mice (all on p62-KO and ApoE-KO background) were fed a western diet for 2 months to develop lesions—exact genotypes are provided at the top of the graph. Atherosclerotic plaque burden was quantified by computer image analysis of Oil Red O-stained aortic root sections with representative Oil Red O-stained aortic roots shown on right (scale bar, 0.4 mm; statistical significance of differences was calculated using Mann–Whitney U-test). **(b)** Macrophage content in aortic root sections was analysed by immunofluorescence staining using an antibody against MOMA-2 ($n \geq 11$ mice per group). **(c)** Apoptotic, necrotic core and combined apoptotic–necrotic core areas of aortic root sections were determined by quantification of TUNEL immunofluorescence staining, acellular areas or a combination of the two, respectively ($n \geq 11$ mice per group). **(d)** Immunofluorescence images of p62-KO and p62-KO/TFEB-TG macrophages treated with or without cholesterol crystals for 24 h and stained using DAPI (nuclei), and antibodies against polyubiquitinated proteins (FK-1) and p62 (scale bar, 5 μm). **(e)** Quantification of average ubiquitin intensity per cell from immunofluorescence experiment described in **d** (numbers of quantified cells are shown under each bar). **(f)** p62-KO and p62-KO/TFEB-TG macrophages were incubated with cholesterol crystals and the per cent of caspase 3/7-positive cells was quantified in three independent experiments (numbers of quantified cells are shown under each bar). **(g)** p62-KO and p62-KO/TFEB-TG macrophages were treated with LPS + CC for 24 h or with LPS followed by ATP for 3 h. Culture media were assayed for IL-1β by ELISA ($n = 3$ independent wells for each treatment). For all graphs, data are presented as mean ± s.e.m. *$P < 0.05$, ***$P < 0.001$, NS: not significant, two-tailed unpaired t-test, except **a**.

course after bolus injection, trehalose clearly peaks between 30 and 60 min (up to a concentration of ∼10 mM) and then largely redistributes or is cleared by 2 h (Fig. 7a). We further determined the kinetics of trehalose in relevant target tissues (aorta, heart and spleen) by mass spectrometry, and found more sustained tissue levels with trehalose being detectable for up to 4 h (Fig. 7b). Trehalose levels in the aorta were also corroborated by a colorimetric assay (Supplementary Fig. 7a). On the cellular level, we also found that trehalose is indeed capable of being taken by macrophages in a concentration-dependent manner (Fig. 7c).

On the basis of this analysis, we chose reasonable doses of trehalose at 1/10th and 1/100th (that is, 1 mM and 100 μM) of the observed peak serum dose to conduct all of our culture analyses. Using macrophages from GFP-LC3 mice, we first conducted live-cell imaging to quantify autophagosome formation upon short-term trehalose exposure. Trehalose (1 mM but not 100 μM) was able to minimally elevate autophagy within minutes that persisted at low levels for up to 1 h but was clearly not as potent as nutrient starvation (PBS) or the autophagosome buildup seen with the lysosomal inhibitor bafilomycin (Fig. 7d,e, Supplementary

Fig. 7b–g and Supplementary Movies 7–10). In contrast, macrophages treated with trehalose for longer periods (3, 6 and 12 h) displayed more potent autophagy-inducing effects (Fig. 7f and Supplementary Fig. 7h). The need for longer incubation times raised the possibility that trehalose's effects might also relate to transcriptional activation of autophagy and induction of autophagy–lysosomal biogenesis. Indeed, beginning at 3 h, trehalose significantly induced transcripts for a variety of autophagy and lysosomal genes (Fig. 7g) and concomitantly induced their protein expression (Fig. 7h and Supplementary Figs 7i and 11b). Moreover, trehalose was still able to induce some transcriptional activation of autophagy–lysosomal genes even at 100 μM, a dose at which we

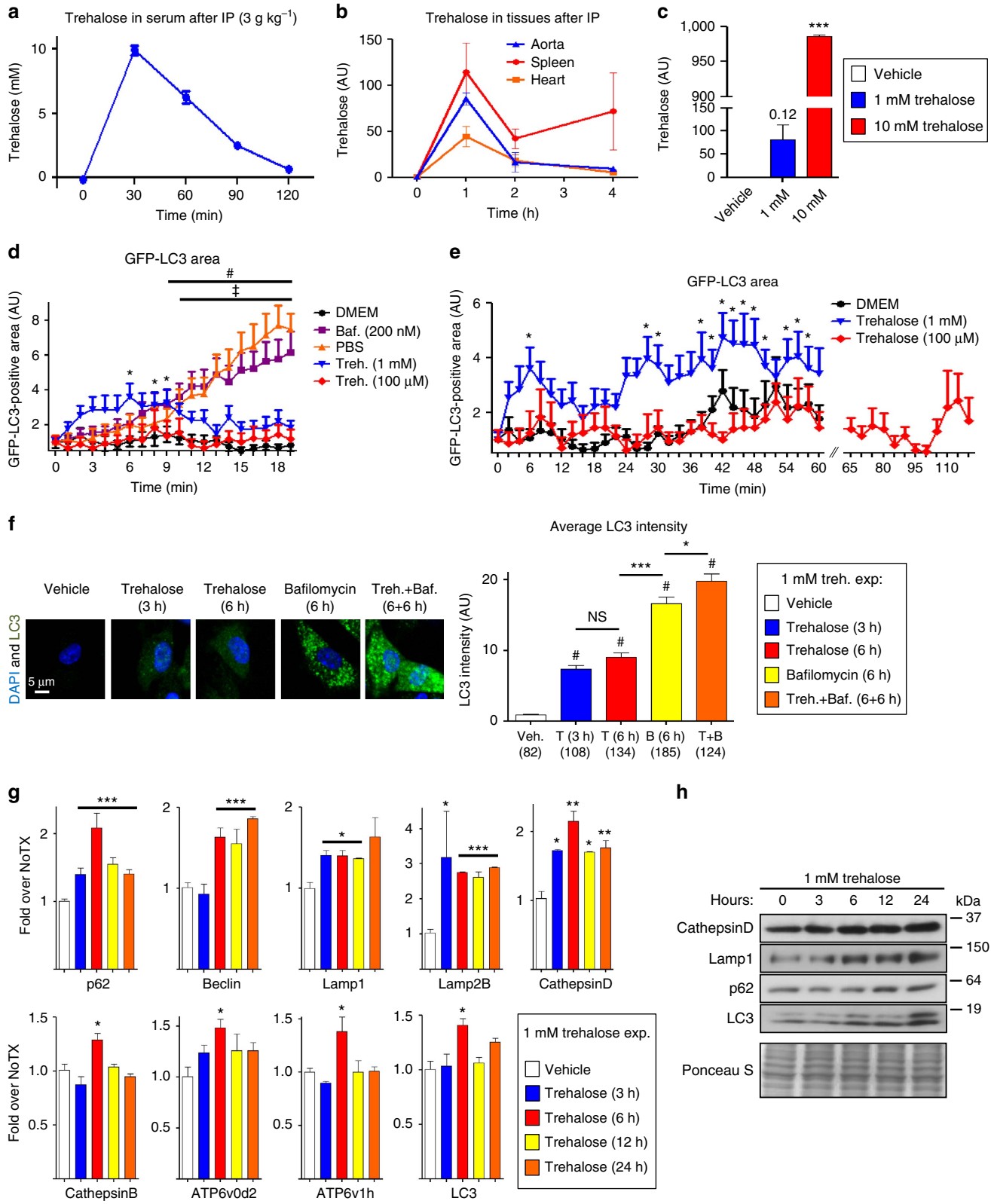

observed no autophagy induction during the short time course experiments (Supplementary Fig. 7j), suggesting a more potent role for trehalose in autophagy–lysosomal biogenesis. In this regard, we also tested whether relatively short exposure to trehalose could potentially sustain autophagy for longer durations. Interestingly, macrophages transiently incubated with trehalose for 3 h still maintained elevations in the autophagy markers LC3 and p62 for an additional 9 h even in the absence of additional trehalose exposure (Supplementary Figs 7k and 12c). These sets of experiments suggested that trehalose can promote a sustained autophagy–lysosomal biogenesis programme in macrophages with peak effect on the order of 6–12 h.

**Trehalose drives and functionally mimics TFEB activation.** Supporting a role for trehalose in autophagy–lysosomal biogenesis and TFEB activation, trehalose was able to increase both TFEB expression at both 1 mM and 100 μM doses as well as TFEB nuclear translocation, a surrogate marker for TFEB activation, and its transcriptional effects (Fig. 8a,b and Supplementary Fig. 11c). Time course experiments conducted over 24 h suggested maximal trehalose-induced TFEB expression to occur at approximately the 6-h time point coinciding with the transcriptional activation of the autophagy–lysosomal genes (Supplementary Figs 8a and 12d). Since TFEB is a member of the MiTF transcription factor family (which includes MiTF and TFE3) and those transcription factors have also been previously implicated as inducers of autophagy–lysosomal biogenesis in different tissues[31,32], we also tested the effects of trehalose on MiTF and TFE3. Interestingly, trehalose had similar stimulatory effects on TFE3 (as gauged by increased TFE3 transcripts and nuclear translocation) but not MiTF (Fig. 8c–e and Supplementary Figs 8b–e and 11d).

Because trehalose has the capacity to induce transcriptional activation of autophagy and autophagy–lysosomal biogenesis in macrophages, we sought to assess its potential to recapitulate the functional benefits of macrophage TFEB activation we had observed in the mϕTFEB-TG system. Indeed, trehalose reduced atherogenic lipid-induced p62-enriched polyubiquitinated protein accumulation as assessed by both immunoblot and IF (Fig. 8f–i and Supplementary Figs 8f and 11e). Trehalose also concomitantly blunted macrophage apoptosis and IL-1β secretion in assays similar to those we conducted for the TFEB-overexpressing macrophages above (Fig. 8j,k).

**Trehalose induces autophagy *in vivo* and is atheroprotective.** To evaluate whether trehalose has autophagy- and TFEB-inducing properties directly in the plaque, we fed a cohort of GFP-LC3 mice (on a pro-atherogenic ApoE-null background) a western diet for 6 weeks to develop atherosclerotic lesions followed by an additional 2 weeks of trehalose administration. A combined strategy of i.p. (2 g kg$^{-1}$ body weight, daily) and oral (3% w/v, *ad libitum*) trehalose administration was pursued since similar dosing strategies have been used in prior trehalose studies of neurodegenerative/protein aggregation disorders[22,26,28,29]. IF quantification of aortic root atherosclerotic lesions revealed both increased GFP-LC3 fluorescence (Fig. 9a,b), together with enhanced TFEB staining and TFEB nuclear localization (Fig. 9c–e and Supplementary Fig. 9a,b). Splenic macrophages derived from trehalose-treated mice also showed increased transcripts for a variety of autophagy and lysosomal genes (Supplementary Fig. 9c). These data indicate trehalose's ability to induce autophagy as well as TFEB activation *in vivo* in agreement with our cultured macrophage data and provided the impetus to conduct a longer-term atherosclerosis study.

A large cohort of pro-atherogenic ApoE-null mice were placed on a western diet, treated with vehicle, trehalose or sucrose (a similar non-reducing disaccharide), and atherosclerosis was quantified at the level of the aortic root (Fig. 9f, details are this study). Trehalose administration had no effects on serum cholesterol levels or other common metabolic parameters (Fig. 9g and Supplementary Fig. 9d,e). Lesion quantitation revealed that trehalose-treated mice were significantly protected from atherosclerosis compared to vehicle (Fig. 9h). Furthermore, the administration of sucrose had no effect and was, in fact, slightly atherogenic (Fig. 9h). Although the non-reducing nature of sucrose makes it the ideal control disaccharide in comparison to trehalose, sucrose is α,β-1,2-linkage of glucose with fructose rather than trehalose's α,α-1,1-linkage of glucose with glucose. Thus, we also administered the reducing disaccharide maltose (α,α-1,4-linkage of glucose and glucose) to ApoE-null mice in order to ascertain the uniqueness of trehalose as an atheroprotective disaccharide. Maltose administration had no detectable effect on atherosclerotic plaque burden or on cholesterol levels (Supplementary Fig. 9f,g).

Finally, our dual i.p. and oral dosing strategy raised the question of which route of administration contributes most significantly to serum trehalose levels and the observed protection from atherosclerosis. We were surprised to discover that i.p. administration of trehalose leads to far greater elevations in serum trehalose than oral dosing (Fig. 9i). Serum trehalose levels rose minimally after oral dosing despite the far greater excess (by weight) provided than i.p. dosing. Thus, to rule-out off-target effects of oral trehalose in affecting atherosclerotic lesion formation (for example, effects of trehalose on the gastrointestinal system or gut microbiota), we conducted an oral-only trehalose experiment in a cohort of ApoE-null

---

**Figure 7 | Trehalose induces autophagy and the transcription of autophagy–lysosomal genes in macrophages.** (**a,b**) Time course of serum and tissue trehalose levels from wild-type mice ($n \geq 4$) after trehalose administration (3 g kg$^{-1}$ i.p.) by colorimetric method and mass spectrometry, respectively. (**c**) Macrophages were treated with vehicle or trehalose at indicated concentrations for 3 h and intracellular trehalose levels measured by mass spectrometry ($n = 2$ independent wells). (**d**) GFP-LC3-expressing macrophages were imaged live every 30 s while being incubated in DMEM (control) ± bafilomycin (200 nM), PBS (starvation) or trehalose (1 mM, 100 μM) for 20 min. GFP-LC3-positive area was quantified ($n \geq 10$ cells for each treatment) and plotted relative to 0 min. No significant difference seen for DMEM and 100 μM trehalose treatments. Trehalose (1 mM), bafilomycin and PBS significance is demarcated by *, # and ‡, respectively ($P < 0.05$ for all cases). (**e**) Protocol as in **d** but macrophages were imaged live for 1 h (DMEM and 1 mM trehalose) or 2 h (100 μM trehalose). No significant difference seen for DMEM and 100 μM trehalose treatments. *$P < 0.05$ for 1 mM trehalose treatment time points. (**f**) Wild-type macrophages were incubated with trehalose (1 mM; 3 and 6 h), bafilomycin (200 nM; 6 h) or both (6 h trehalose pretreatment and 6 h co-incubation) and stained with LC3 antibody and DAPI. Representative images are shown at left and quantification of average LC3 intensity with each condition at right (number of cells under each bar). #shows significant difference compared to vehicle (using analysis of variance (ANOVA) followed by Tukey's multiple comparison test; scale bar, 5 μm). (**g**) Wild-type macrophages were treated with 1 mM trehalose for indicated time points and transcripts of autophagy–lysosomal genes detected by qPCR ($n \geq 3$ independent wells for each gene). (**h**) Western blot analysis of Cathepsin D, Lamp1, p62 and LC3 in macrophages after 1 mM trehalose treatment for indicated times. Ponceau S staining used as loading control. For graphs in **c**,**f**,**g**, data are presented as mean ± s.e.m. *$P < 0.05$, **$P < 0.01$, ***$P < 0.001$, NS: not significant, two-tailed unpaired *t*-test compared to zero time point or vehicle, except **f**.

mice over 8 weeks of concomitant western diet feeding. Oral trehalose had no observable effects on either atherosclerosis or serum cholesterol (Fig. 9j and Supplementary Fig. 9h).

**Trehalose reduces atherosclerosis through autophagy and p62.** Similar to mɸTFEB-TG mice, the effects of trehalose *in vivo* and on cultured macrophages *in vitro* demonstrate this unique disaccharide's ability to increase macrophage autophagy and autophagy–lysosomal biogenesis, to increase aggrephagy and the clearance of p62-enrich protein aggregates, to decrease macrophage apoptosis and the pro-inflammatory cytokine IL-1β and to an overall reduction in atherosclerosis. We set out to determine whether trehalose's autophagy-p62-mediated effects are mechanistically linked to the observed reduction in atherosclerosis by administering trehalose to mɸATG5-KO

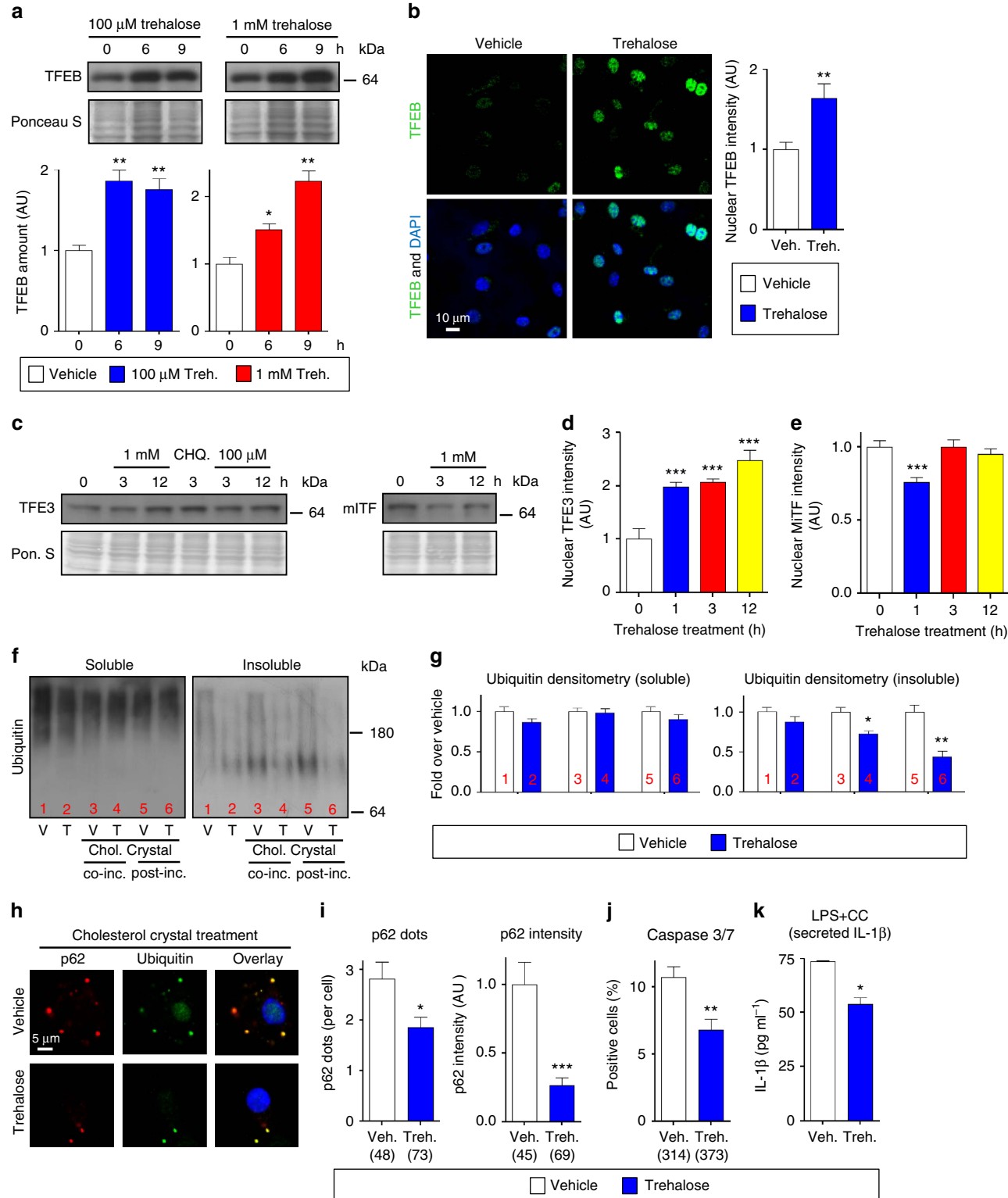

and p62-KO mice (both on pro-atherogenic ApoE-null backgrounds).

Consistent with its specificity towards macrophage autophagy, trehalose was no longer able to reduce atherosclerosis in mɸATG5-KO mice (Fig. 10a and Supplementary Fig. 10a–c). In addition, trehalose was no longer able to reduce the number of p62-enriched protein aggregates or blunt the degree of apoptosis when subjected to atherogenic lipids (Fig. 10b,c). Interestingly, trehalose's blunting effect on IL-1β secretion was independent of autophagy, again suggesting an independent mechanism that appears to have no physiological significance for atherosclerotic plaque formation (Fig. 10d). Finally, abrogation of the atheroprotective effects of trehalose, its ability to reduce polyubiquitinated protein aggregates and macrophage apoptosis, and apparent irrelevance of IL-1β reduction to these phenotypes were entirely recapitulated in p62-KO mice (Fig. 10e–h and Supplementary Fig. 10d–f).

## Discussion

In this study, we provide both genetic and pharmacological evidence supporting a protective role for the stimulation of the macrophage autophagy–lysosomal biogenesis programme in atherosclerosis. Using the common marker of autophagy progression (LC3) and the autophagy chaperone (p62), we first show a progressive autophagy dysfunction in the developing atherosclerotic plaque. We then utilize a macrophage-specific TFEB overexpression mouse model to activate the predominant transcription factor mediating autophagy–lysosomal biogenesis in macrophages and note its ability to reprogramme plaque macrophages to co-express LC3 and p62. We show that TFEB overexpression is able to increase macrophage autophagy both in vitro and in vivo, and it has functional benefits by increasing aggrephagy of cytotoxic p62-enriched protein aggregates concomitant with blunting of lipid-induced macrophage apoptosis and IL-1β production. This in turn leads to reduction of atherosclerotic plaque burden and the development of less complex lesions with reduced plaque macrophage apoptosis and necrotic cores. Importantly, using models of autophagy- and autophagy chaperone deficiency (ATG5-KO and p62-KO) we show the specificity of macrophage TFEB's atheroprotective action. The absence of ATG5 or p62 abrogates TFEB's ability to reduce atherosclerosis, plaque complexity, macrophage apoptosis and aggrephagy of p62-enriched protein aggregates.

In an attempt to leverage the beneficial therapeutic effects of stimulating macrophage autophagy–lysosomal biogenesis, we describe the utility of the natural sugar trehalose with previously described but mechanistically unknown autophagy-inducing properties. We show that therapeutically relevant doses of trehalose are able to stimulate macrophage autophagy, autophagy–lysosomal biogenesis and TFEB both in vitro and in vivo. Mirroring TFEB action, trehalose increases aggrephagy of cytotoxic p62-enriched protein aggregates, blunts lipid-induced macrophage apoptosis and IL-1β production, and reduces atherosclerotic plaque burden. Similarly, these effects are abrogated in models of autophagy- and autophagy chaperone deficiency (ATG5-KO and p62-KO). This comprehensive evaluation of both genetic and pharmacological methods of enhancing macrophage autophagy–lysosomal biogenesis demonstrates the utility of this approach in cardiovascular therapeutics (overview shown in Fig. 10i).

Several unanswered questions remain. First, there is an ever-expanding list of degradative functions for the autophagy–lysosomal system spanning both protein and organelle turnover. In atherosclerosis, the main three proposed mechanisms are dampened inflammasome/IL-1β activation, accelerated removal of protein aggregates and increased cholesterol efflux. We have focused on two of these mechanisms in the functional evaluation of the TFEB and trehalose. Our data support a critical role for the aggrephagy of p62-enriched protein aggregates. We have recently shown that the formation and autophagic processing of p62-enriched inclusion bodies is a critical process by which plaque macrophages mitigate atherosclerotic plaque burden[6]. This was evaluated using detailed phenotyping of macrophage ATG5-, p62- and dual ATG5/p62-deficient mice. Now, we demonstrate that the autophagy-inducing and atheroprotective effects of TFEB or trehalose are critically dependent on aggrephagy, resulting in removal of p62-enriched inclusion bodies and a reduction in macrophage apoptosis. Surprisingly, blunted IL-1β release does not appear to be a mechanistic contributor as the IL-1β-reducing effects of both TFEB and trehalose are independent of ATG5 or p62. In many neurodegenerative disorders where protein aggregation is pathogenic and now most recently in atherosclerotic plaques, it is evident that corralling insoluble protein aggregates into p62-enriched inclusions and clearing these inclusions via autophagy is an important response to prevent cytotoxicity and apoptosis. However, the exact mechanisms by which insoluble protein aggregate accumulation can cause apoptosis remain unknown. Future work delineating the protein and organelle composition of these p62-enriched inclusion bodies could give clues to this link.

Although our work implicates the autophagic clearance of pro-apoptotic and pro-inflammatory inclusion bodies as a major

**Figure 8 | Trehalose induces TFEB nuclear localization and protects from atherogenic lipid-induced protein aggregation and related sequelae of apoptosis and inflammasome activation.** (**a**) Western blot analysis of TFEB in macrophages after trehalose treatment for indicated times. Ponceau S staining is shown as loading control and densitometric quantification from three separate experiments is shown below. (**b**) TFEB nuclear localization is analysed by immunofluorescence staining after trehalose treatment and graphed as nuclear TFEB intensity ($n \geq 40$ cells per group, scale bar, 10 μm). (**c**) Western blot analysis of other MiTF transcriptional family members (TFE3 and MiTF) after trehalose or chloroquine (CHQ, 10 μM) treatments for indicated doses and times. Ponceau S staining is shown as loading control. (**d,e**) TFE3 and MiTF nuclear localization was analysed by immunofluorescence staining after trehalose treatment for indicated times and quantified by the intensity of nuclear staining ($n \geq 500$ cells per group). (**f**) Western blot analysis of polyubiquitinated proteins (FK-1 antibody) in detergent-soluble and detergent-insoluble lysate fractions of vehicle (V) or trehalose (T) treated wild-type macrophages. Lanes 3 and 4 were either vehicle or trehalose pretreated for 3 h, and then co-treatment with cholesterol crystals is performed for 12 h. Lanes 5 and 6 were cholesterol crystal-treated for 6 h and then either treated with vehicle or trehalose alone for another 6 h. (**g**) Densitometric quantification of **f** from three similar separate experiments. (**h**) Immunofluorescence images of wild-type macrophages after indicated treatments using DAPI and antibodies against polyubiquitinated proteins (FK-1) and p62 (scale bar, 5 μm). (**i**) Graphs represent average p62/ubiquitin + dot numbers and average p62 intensity per cell for immunofluorescence images in **h** (numbers of quantified cells are shown under each bar). (**j**) Wild-type macrophages were co-incubated with cholesterol crystals and trehalose (or vehicle); the per cent of caspase 3/7-positive cells was quantified in three independent experiments (numbers of quantified cells are shown under each bar). (**k**) Wild-type macrophages were treated as indicated and cell culture media were assayed for IL-1β by ELISA ($n = 3$ independent wells for each treatment). For all graphs data are presented as mean ± s.e.m. *$P < 0.05$, **$P < 0.01$, ***$P < 0.001$, two-tailed unpaired t-test compared to zero time point or vehicle treatment group.

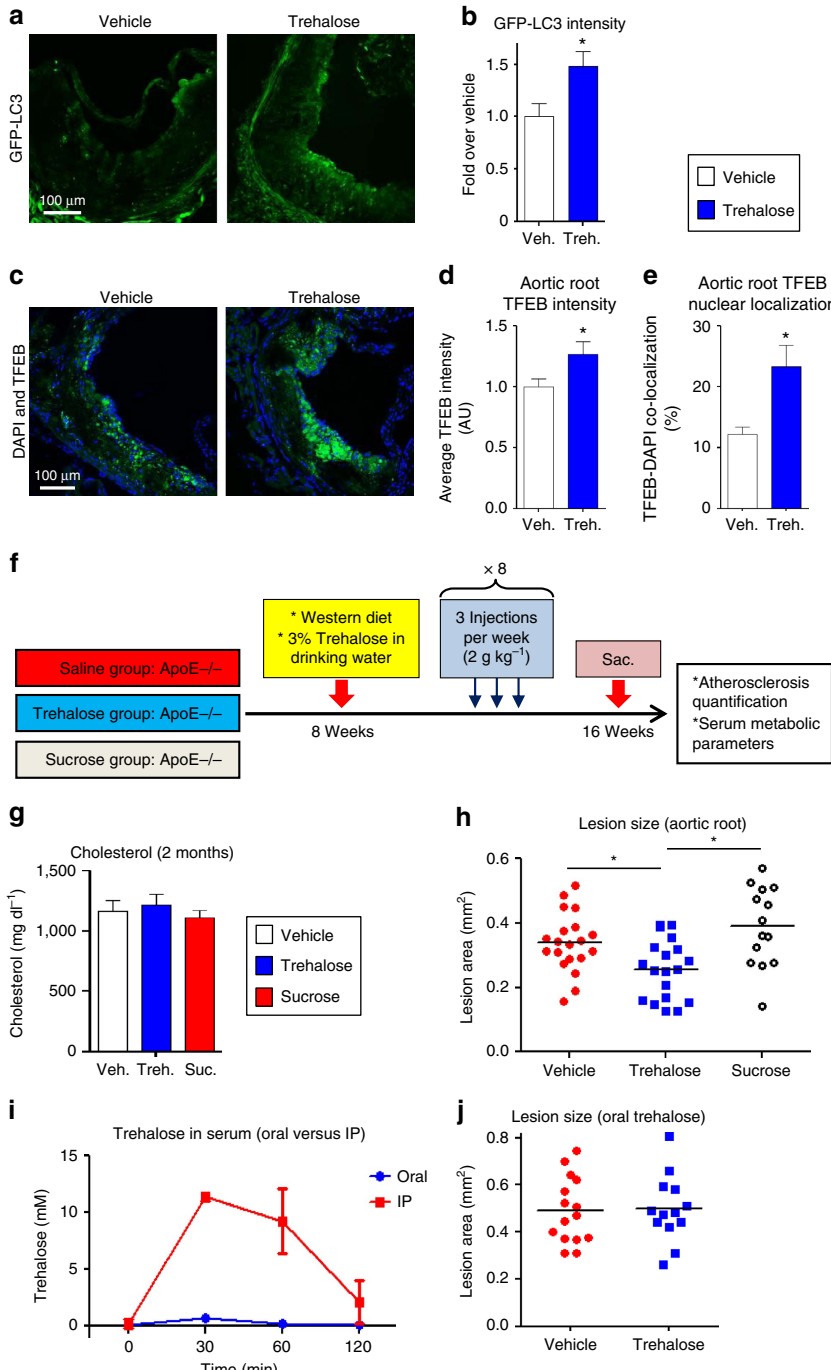

**Figure 9 | Trehalose administration in mice is atheroprotective.** (**a,b**) GFP-LC3 mice (ApoE-KO background) were fed western diet for 2 months and administered vehicle or trehalose (2 g kg$^{-1}$ given five times per week i.p.) in the final 2 weeks of diet ($n = 4$ mice per group). Representative GFP fluorescence (scale bar, 100 μm; **a**) and quantification of GFP intensity in aortic roots by confocal microscopy ($n = 4$ mice per group; **b**) is shown. (**c–e**) TFEB intensity and nuclear localization in aortic roots of the same cohort used in **a** was detected by immunofluorescence. Shown are (**c**) representative aortic root TFEB staining (scale bar, 100 μm), (**d**) average aortic root TFEB intensity; and (**e**) TFEB-DAPI co-localization. (**f**) Diagram summarizing experimental protocol and mice cohorts used for *in vivo* assessment of trehalose in atherosclerosis. Mice were fed a western diet for 2 months while being administered either vehicle, trehalose or sucrose (disaccharides given both i.p. 2 g kg$^{-1}$ for three times per week and orally 3% *ad libitum* in drinking water). (**g**) Serum cholesterol levels at 2 months of western diet ($n = 7$ mice per group). (**h**) Quantification of atherosclerotic plaque burden by computer image analysis of Oil Red O-stained aortic root sections in the experiment summarized in **f** (statistical significance of differences was calculated using Mann–Whitney *U*-test). (**i**) Serum trehalose levels in wild-type mice ($n = 4$ per group) after administration of trehalose (3 g kg$^{-1}$) either by i.p. injection or oral gavage at indicated time points. (**j**) Quantification of atherosclerotic plaque burden by computer image analysis of Oil Red O-stained aortic root sections in mice fed 2 months of western diet while being administered only oral trehalose (statistical significance of differences was calculated using Mann–Whitney *U*-test). All data are presented as mean ± s.e.m. *$P < 0.05$, two-tailed unpaired *t*-test except **h,j**.

mechanism by which induction of macrophage autophagy–lysosomal biogenesis ameliorates atherosclerosis, it is important to consider other contributing factors to the observed plaque reduction. Both mφTFEB-TG mice and trehalose-treated mice developed smaller lesions because of a combined decline in lesion complexity (necrotic core and apoptosis) and macrophage

burden. Factors that mitigate lesional macrophage accumulation include effects on both ins and outs (that is, monocyte numbers/infiltration versus macrophage clearance via efferocytosis). In the experiments we performed on thioglycollate-elicited peritoneal macrophage and splenic macrophages, we did not observe changes in macrophage numbers, suggesting that overall

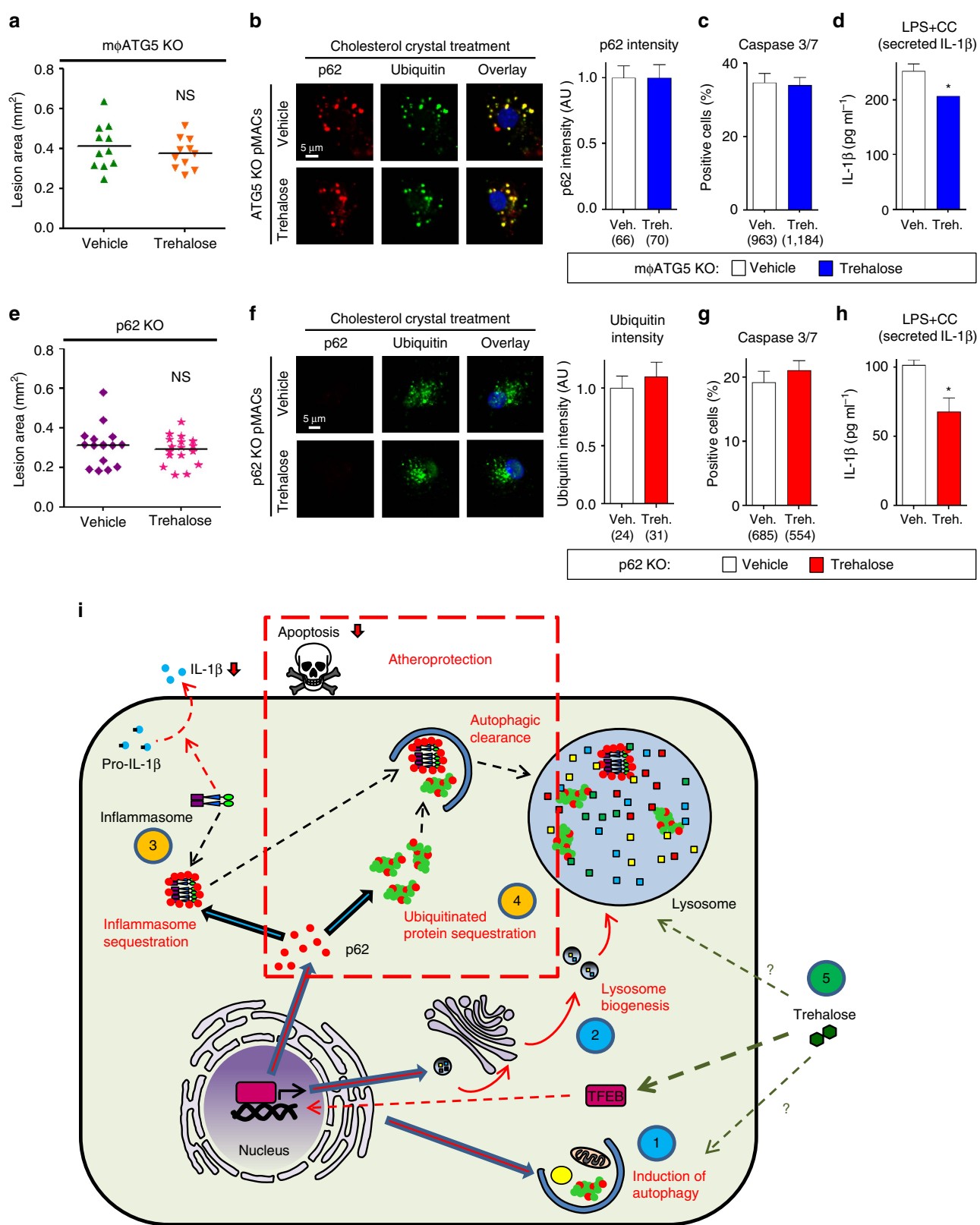

monocyte response and tissue infiltration is likely normal. However, given reports linking components of the autophagy machinery to phagocytosis[4,33], it would be interesting to investigate the additional effects of TFEB, trehalose and autophagy–lysosome biogenesis on phagocytic processes generally and efferocytosis in particular.

This is the first report demonstrating the use of the natural sugar trehalose in the treatment of atherosclerosis. We corroborate previous studies conducted in other disease systems that trehalose has potent autophagy-inducing properties with an overall beneficial disease-modifying profile. Compared to many other trehalose studies, we have gone further by actually demonstrating the autophagy dependence of trehalose both *in vitro* and *in vivo*. Trehalose's ability to clear p62-enriched inclusion bodies in macrophages further shows that trehalose's beneficial effects are specifically dependent on p62 and aggrephagy. Interestingly, this appears to be mechanistically similar to prior reports of trehalose in neurodegenerative disorders (for example, Huntington's, Parkinson's and amyotrophic lateral sclerosis), which reported its ability to induce the aggrephagy of neuronal cytoplasmic protein inclusions[22,24,26].

Despite the broad ability of trehalose to induce autophagy in disparate model systems, the precise mechanism by which this occurs is still unknown. Our interrogation of the timing of this autophagy induction by live-cell imaging and assessment of the autophagy–lysosomal biogenesis response suggests that a prominent effect of trehalose action is the stimulation of TFEB and broad transcriptional activation of autophagy and lysosomal genes. How is this disaccharide capable of inducing TFEB nuclear translocation and its transcriptional activation? It is intriguing to note that trehalose is not inherently cell-permeable and is instead likely to be taken up by fluid-phase endocytosis[34]. This would indicate that trehalose accumulation in the endosome–lysosome system and modulation of its function is a distinct possibility. In addition, compared to most disaccharides and particularly non-reducing disaccharides, trehalose is relatively resistant to acid hydrolysis favouring its lysosomal retention in an undegraded form[20]. Since modulation of lysosome function and pH are potent triggers for TFEB nuclear translocation (for example, TFEB is activated by the lysosomotropic acid suppressors chloroquine and bafilomycin)[35,36], it would be interesting to further study this trehalose–lysosome–TFEB dynamic. It should be noted that our evaluation of the other members of the MiTF/TFE transcription factor family (namely TFE3 and MiTF) illustrates trehalose's ability to also trigger TFE3 but not MiTF nuclear translocation. As all three family members (in particular TFEB and TFE3) have been implicated in upregulation of autophagy and lysosome genes[30,31], future work using macrophage-specific knockout mice for each and combinations of these transcription factors will be

needed to determine the specificity of trehalose on macrophage autophagy–lysosomal biogenesis and effects on atherogenesis.

In addition, systemic administration of an autophagy-inducing compound could potentially have off-target effects by inciting cellular degradation in normal tissue, although we are not aware of overt deleterious effects being noted for trehalose. Hyperactive autophagy has been a proposed alternative to apoptotic cell death (that is, autophagy-mediated cell death or most recently termed autosis)[37]. Rather, our data suggest that the level of autophagy and autophagy–lysosomal biogenesis induced by trehalose is protective against lipid-induced macrophage death *in vitro* and in the plaque.

Despite the prospect of a safe and natural sugar as a treatment for cardiovascular disease, we did not observe protection from atherosclerosis with oral-only dosing of trehalose. Oral trehalose resulted in minimal elevations of serum trehalose levels in contrast to the robust levels seen with intraperitoneal administration. The likely reason for this dramatic first-pass effect on ingested trehalose is the presence of mammalian trehalase, the trehalose-specific glycosidase present in highest amounts in the gastrointestinal tract, kidney and liver of all mammals spanning mice to humans[20,38]. Thus, future evaluations of trehalose in animal models and potentially clinical trials of its cardiovascular efficacy will need to largely consider parenteral formulations or investigate methods to pharmacologically inhibit trehalase while administering oral trehalose. This can be directly tested by determining whether oral trehalose absorption can be augmented and lead to lower atherosclerosis in trehalase-deficient mice. It is interesting to note that a recent human genetics study described single nucleotide polymorphisms in the trehalase gene (*TREH*), which resulted in lower levels of trehalase activity and lower diabetes risk in Pima Indians[39]. An evaluation of the *TREH* locus in large-scale cardiovascular genetics cohorts will also be insightful.

## Methods

**Animals.** Animal protocols were approved by the Washington University Animal Studies Committee. All mice used in this study were on C57BL/6 J background (>N7). Mice with TFEB-TG and ATG5 deficiency (ATG5-KO) were as previously described[13,40]. These mice were crossed with Cre-recombinase transgenic mice under the control of the Lysozyme-M promoter to generate macrophage-specific ATG5-KO (mφATG5-KO) and TFEB-TG (mφTFEB-TG) mice[3,12]. p62-KO and GFP-LC3 mice were as described previously[6,41]. Crosses between mφTFEB-TG, mφATG5-KO, p62-KO, GFP-LC3 and ApoE-KO mice generated mice with mφTFEB-TG, mφATG5-KO, p62-KO, mφTFEB-TG/ATG5-KO, mφTFEB-TG/p62-KO, mφTFEB-TG/GFP-LC3 or littermate controls on an ApoE-KO background. TFEB-TG, p62, ATG5, Cre, GFP-LC3 and ApoE genotyping was performed using standard PCR techniques (Supplementary Table 1)[6,13,40–42]. Mice housed in a specific pathogen-free barrier facility were weaned at 3 weeks of age to a standard mouse chow, providing 6% calories as fat. For *in vivo* experiments, male mice were started on a western-type diet containing 0.15% cholesterol and

**Figure 10 | Atheroprotective effects of trehalose are dependent on macrophage autophagy and p62.** (**a,e**) mφATG5-KO (**a**) or p62-KO mice (**e**; all on ApoE-null background) were fed a western diet for 2 months with concurrent vehicle or trehalose administration (2 g kg⁻¹ given three times per week i.p. and 3% *ad libitum* in drinking water). Graphs represent quantification of Oil Red O-stained atherosclerotic plaques at the level of aortic root (statistical significance of differences was calculated using Mann–Whitney *U*-test). (**b,f**) Immunofluorescence images of ATG5-KO (**b**) or p62-KO (**f**) macrophages using DAPI and antibodies against polyubiquitinated proteins (FK-1) and p62 (scale bar, 5 μm). Average p62 intensity (**b**) or ubiquitin intensity (**f**) per cell and number of cells is shown. (**c,g**) ATG5-KO (**c**) or p62-KO (**g**) macrophages were co-incubated with cholesterol crystals and trehalose (or vehicle). Per cent of caspase 3/7 + cells were quantified in three independent experiments (number of cells shown under each bar). (**d,h**) ATG5-KO (**d**) or p62-KO (**h**) macrophages were treated as indicated and cell culture media assayed for IL-1β by ELISA (n = 3 independent wells for each treatment). Data are presented as mean ± s.e.m. *P < 0.05, NS: not significant, two-tailed unpaired *t*-test except **a,e**. (**i**) Graphical summary of the benefits of harnessing macrophage autophagy–lysosomal biogenesis in atherosclerosis: TFEB overexpression in macrophages initiates its nuclear localization and autophagy–lysosomal biogenesis (1 and 2). This activation modulates several downstream mechanisms in macrophages that contribute to a reduction in atherosclerosis (3 and 4), such as decreased IL-1β secretion (3), p62-dependent polyubiquitinated protein sequestration and autophagic clearance and decreased apoptosis (4). Our data largely implicate the atheroprotective effects of TFEB to be dependent on the aggrephagy of p62-enriched inclusion bodies and associated reductions in apoptotic cell death (4). The disaccharide trehalose is an inducer of macrophage autophagy and autophagy–lysosomal biogenesis and reduces atherosclerosis by recapitulating these TFEB-induced pathways (5).

42% calories as fat (TD 88137, Harlan) at ∼2 months of age. Unless otherwise noted, experimental groups that were administered various disaccharides (that is, trehalose, sucrose or maltose) were injected i.p. (2 g kg$^{-1}$, three times per week) while drinking water was supplemented at 3% w/v. For *in vitro* experiments, both male and female mice aged between 2 and 6 months were used. In each experiment only samples from the same sex and similar age were compared.

**Human carotid samples.** Carotid tissue specimens were obtained from consenting de-identified patients from the Institutional Review Board-approved Washington University School of Medicine Vascular Surgery biobank. The material removed at the time of CEA was directly placed in cold saline and kept on ice during the transport to the laboratory. Samples were then rapidly dissected into maximally diseased atherosclerotic regions and adjacent minimally diseased regions, and placed in a tissue-freezing medium for sectioning and subsequent immunostaining.

**Macrophage culture and treatment.** Standard techniques were used to isolate thioglycollate-elicited peritoneal macrophages[43]. Briefly, mice were injected with 4% sterile thioglycollate media (Sigma, T9032) i.p., and 4 days later, peritoneal macrophages were isolated, washed, counted and plated (DMEM with 10% fetal bovine serum). Following treatments were performed on macrophages: cholesterol crystals (500 mg ml$^{-1}$), chloroquine (10 μM; Sigma, C6628), bafilomycin (100 nM; Sigma, B1793), lipopolysaccharide (100 ng ml$^{-1}$; InvivoGen), oxLDL (50 μg ml$^{-1}$), ATP (5 mM; Sigma, A2383) and/or trehalose (100 μM and 1 mM; Sigma, T0167). Treated cells were either harvested at various times for mRNA or protein isolation using standard techniques or were fixed with 4% paraformaldehyde for IF microscopy. Cholesterol crystals were generated by subjecting cholesterol powder (Sigma, C8667) to an ethanol precipitation technique[44,45]. Cholesterol was dissolved in ethanol (50 mg ml$^{-1}$) at 65 °C and crystallized by incubation at −20 °C overnight. Crystals were collected by centrifugation at 3,000 r.p.m., and the remaining ethanol was cleared by subsequent ddH$_2$O washes. Oxidized low-density lipoprotein (LDL) was generated by Cu$^{2+}$ oxidation of LDL (Sigma, L7914)[46]. LDL (500 μg ml$^{-1}$) was dialysed into EDTA-free PBS, CuSO$_4$ was added to LDL solution to a final concentration of 10 μM and the solution was incubated at 37 °C for 6 h. Reaction was terminated by dialysing the LDL solution against 2 mM EDTA for 4 h.

**FACS analysis.** FACS analysis for the assessment of autophagy and lysosomal markers was performed on cultured macrophages, mouse spleens and aortas. Macrophages were plated (Greiner 665102), adherent cells were collected with CellStripper (Corning 25056), fixed in 4% paraformaldehyde and stained with phycoerythrin-conjugated LAMP1 antibody (BioLegend 121612, 1:200). For splenic macrophages, spleens were minced, filtered to single-cell suspensions using a 70 μm cell strainer, subjected to red blood cell lysis, and stained with Pacific Blue-conjugated CD45 (BioLengend 103126, 1:200), fluorescein isothiocyanate-conjugated F4/80 (BioLengend 123108, 1:200), PerCP-Cy5.5-conjugated CD11b (BioLengend 101228, 1:200) and phycoerythrin-conjugated LAMP1 (BioLengend 121612, 1:200) antibodies. Aortas (extending from the aortic root to the abdominal aorta at the level of the renal arteries) were dissected after PBS perfusion, cleaned of all surrounding tissues, minced and digested for 60 min at 37 °C in a buffer consisting of RPMI, 2.5 μg ml$^{-1}$ Liberase (Roche 05401127001), 125 μg ml$^{-1}$ DNAse 1 (Sigma D4527) and 0.8 mg ml$^{-1}$ hyaluronidase (Sigma H3506). Single-cell suspensions were collected by passing through a 70-μm cell strainer and then labelled with fluorochrome-conjugated macrophage markers CD45, F4/80 and CD11b (as above). For intracellular staining, cells were additionally fixed with 4% paraformaldehyde, permeabilized with 0.3% saponin and were incubated with p62 (Abcam ab56416, 1:150), LC3 (MBL International PM036, 1:200) and/or Lamp2 (Abcam ab13524, 1:200) primary antibodies, followed by Alexa Fluor 488- or Alexa Fluor 594-conjugated secondary antibodies (Invitrogen A11005, A11007 and A11008; all 1:500). All samples were analysed using the BD Biosciences LSR II flow cytometer and quantified using FlowJo software.

**IF microscopy.** IF imaging of macrophages and frozen-tissue sections was performed using standard techniques[47]. Briefly, cells or tissues were fixed with 4% paraformaldehyde, blocked, permeabilized (1% BSA, 0.2% milk powder, 0.3% Triton X-100 in TBS; pH 7.4) and incubated with antibodies sequentially. For all IF imaging analyses of cultured macrophages, at least two mice of each genotype were used in each experiment and experiments were repeated at least three times. For all imaging analyses of tissue sections, a representative image was chosen and shown after analysis of at least four aortic roots or human samples. The number of analysed cells or tissue sections is indicated in the figure legends. Specificity of staining was tested in control experiments either by omitting primary antibodies or using samples from knockout mice where available. The following primary antibodies were used in 1:250 dilutions: p62 (Progen Biotechnik, GP62-C), polyubiquitinated proteins (FK-1; Enzo Life Sciences, BML-PW8805), MOMA-2 (AbD Serotec, MCA519C), LC3 (MBL, PM036), TFEB (MyBioSource, MBS120432 and Bethyl, A303-673A), MiTF (Thermo Scientific, MS-771), TFE3 (Sigma Life Science, HPA023881) and CD68 (Bio-Rad, MCA1957). Species-specific fluorescent secondary antibodies were obtained from Invitrogen/Life Technologies (1:250). CellEvent Caspase 3/7 Green Detection Reagent (Life Technologies, C10423),

DeadEnd Fluorometric TUNEL System (Promega, G3250), LysoTracker-Red DND-99 (Life Technologies, L7528), DiI-acetylated LDL (Life Technologies, L3484) and DiI-oxidized LDL (Life Technologies, L34358) were used according to the manufacturer's protocol. A Zeiss LSM-700 confocal microscope was used for imaging. Signal over the threshold was quantified using ZEN microscope software (Carl Zeiss Microscopy) after determination of regions of interest and thresholds.

**Live-cell imaging.** For live imaging, macrophages were plated on glass-bottom culture dishes (Mattek Corporation, P35G-1.5-10-C). Cells were imaged using a Nikon A1Rsi Confocal Microscope with Tokai-hit stage-top incubator at 37 °C and 5% CO$_2$. Drugs were added after the first image, and images was captured every 30 s. Image analysis and quantification was performed using ImageJ software. Regions of interest with thresholds were determined and signal over threshold was quantified. Baseline images (image1) were set to unity and other time points were displayed as changes over this baseline.

**Preparation of soluble/insoluble fractions and western blotting.** Cells were lysed in a standard lysis buffer (1% Triton X-100; Cell Signaling, #9803).Where soluble and insoluble fractions are needed, samples were sonicated and centrifuged at 17,000g for 15 min and supernatant was defined as the soluble fraction. The pellet (insoluble fraction) was washed with lysis buffer, suspended in boiling SDS–PAGE loading buffer and sonicated for protein solubilization. Standard techniques were used for protein quantification, separation, transfer and blotting[47]. The following primary antibodies were used: TFEB (Bethyl, A303-673A, 1:1,000), Cathepsin D (gift from Dr Stuart Kornfeld, Washington University, 1:4,000), Lamp1 (Santa Cruz Biotechnology, sc-19992, 1:2,000), p62/SQSTM1 (abcam, ab56416, 1:2,000), LC3 (Novus Biologicals, NB100-2220, 1:1,000), polyubiquitinated proteins (FK-2, Millipore, 04-263, 1:1,000), MiTF (Thermo Scientific, MS-771, 1:1,000) and TFE3 (Sigma Life Science, HPA023881, 1:1,000).

**Analytical procedures and lesion quantification.** ELISA assays were performed as per the manufacturer's protocols to detect secreted IL-1β (R&D Systems, MLB00B) and tumour necrosis factor-α (BD Biosciences, 560478) from macrophage culture media. Metabolites (cholesterol, triglycerides and glucose) were assayed in serum obtained after 6-h fast as per the manufacturer's protocols (Thermo Scientific TR13421, TR22421 and TR15408). For assessment of transcript levels, quantitative RT–PCR was performed. Oligonucleotide primers were designed using the qPrimerDepot database http://primerdepot.nci.nih.gov (Supplementary Table 2). Assays were performed at least in triplicate and ribosomal protein 36B4 was used for normalization.

For trehalose tolerance test, mice were fasted for 6 h and 3 g kg$^{-1}$ trehalose was administered by i.p. or oral gavage. Mice were bled from tail vein at different time points, blood was centrifuged and serum was analysed by using a trehalose test kit as per the manufacturer's protocols (Megazyme, K-TREH).

For aortic root cross-section, hearts were placed in a cryostat mold containing a tissue-freezing medium and were frozen after the perfusion with PBS. Ten-μm-thick sections were taken from the samples beginning just caudal to the aortic sinus and extending into the proximal aorta. Slides were fixed with 4% paraformaldehyde and stained with Oil Red O. Images were taken with EVOS XL Core Cell Imaging system, and Oil Red O-positive regions were quantified using ZEN microscope software (Carl Zeiss Microscopy). For *en face* analysis, fixed aorta was incised longitudinally and pinned flat. Analysed regions included the aortic arch, thoracic aorta and abdominal aorta (spanning from the aortic valve to the bifurcation of the iliac arteries)[48].

**Quantification of trehalose levels by mass spectrometry.** The GC–MS (gas chromatography–mass spectrometry) technique was to quantify trehalose levels in both cells and tissues. The isotope [$^{13}$C] trehalose (Omicron Biochemicals, TRE-002) was used as an internal standard during the sample preparation. Samples were extracted into isopropanol:acetonitrile:water (3:3:2), centrifuged at 18,000g (15 min) and dried under N$_2$ gas. N-Methyl-N-(trimethysilyl) trifluoroacetamide with 10% pyridine in CH$_3$CN was then used to derivatize samples for analysis by GC–MS using Agilent 7890A gas chromatograph interfaced to Agilent 5975C mass spectrometer and HP-5ms gas chromatography column (30 m per 0.25-mm internal diameter per 0.25-μm film coating). A temperature gradient was used starting at 80 °C for 2 min, linearly increased at 10 °C min$^{-1}$, ending at 300 °C for an additional 2 min. Samples were subjected to electron ionization mode using source temperature (200 °C), electron energy (70 eV), emission current (300 μA) and injector/transfer line temperatures (250 °C). Monitored ions for trehalose and [$^{13}$C] trehalose were m/z 361 and m/z 367, respectively.

**Isolation of splenic macrophages.** Dissected spleens were minced and single-cell suspensions prepared by passing the sample through a 70-μm cell strainer. Cells were washed with 1% BSA in PBS and re-suspended in MACS buffer (0.5% BSA, 2 mM EDTA in PBS; 10$^7$ splenic cells in 90 μl) followed by addition of 10 μl of CD11b microbeads (Miltenyi Biotec 130-049-601) for 15 min incubation at 4 °C. Cells were washed with MACS buffer, centrifuged at 500g for 10 min at 4 °C, re-suspended in 500 μl MACS buffer and transferred to an equilibrated MS

separation column (Miltenyi Biotec 130-041-301) located in a MACS magnetic separator (Miltenyi Biotec 130-092-168). The column was washed three times with MACS buffer and, upon removal of MS column from the magnetic field, the bound CD11b$^+$ fraction was collected with a plunger. After centrifugation (500$g$ for 5 min at 4 °C), the cell pellet was used for RNA isolation.

**Lysosomal acid lipase activity assay.** Lysosomal acid lipase activity was measured using the fluorogenic substrate 4-methyl-umbelliferyloleate (100 mg ml$^{-1}$ in DMSO). Control or TFEB-TG macrophages were lysed in buffer (10 mM Tris pH 8; 50 mM NaCl; 1% Triton X-100) containing protease inhibitors and protein concentrations were measured using the BCA reagent (Pierce, #23225). A mixture of 4-methyl-umbelliferyloleate substrate, assay buffer (200 mM sodium acetate pH 5.5) and cell lysate (at 1:3:0.2 ratio, 200 μl final volume) was incubated for 30 min at 37 °C and the reaction stopped with 1 M Tris pH 8 (100 μl). Fluorescence intensity was recorded via fluorometric reading (excitation 360 nm/emission 460 nm).

**Lipid extraction.** Lipids were extracted by using a modification of the Folch method[49]. Briefly, cells were homogenized in cold 2:1 chloroform:methanol, and then centrifuged at 12,000 r.p.m. for 10 min at 4 °C. A 15 μl aliquot from the lower organic phase was isolated and evaporated under nitrogen. Triglyceride content was assayed as per the manufacturer's protocols (Thermo Scientific TR22421).

**Statistical analyses.** Statistical significance of differences was calculated using the Student's unpaired $t$-test or analysis of variance (for multiple groups), followed by either Dunnett's test (when multiple groups are compared with a single control) or Tukey's multiple comparison test for parametric data or the Mann–Whitney $U$-test for non-parametric data (atherosclerosis quantitation). Graphs containing error bars show the mean ± s.e.m. Statistical significance is represented as follows: *$P < 0.05$, **$P < 0.01$, ***$P < 0.001$, NS, not significant.

**Data availability.** All the other data that support the findings of this study are available from the corresponding author upon reasonable request.

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

## Acknowledgements

We thank Ms Joan Avery of the Washington University Center for Cardiovascular Research for technical assistance. This work was supported by K08 HL098559, R01 HL125838, the Washington University Diabetic Cardiovascular Disease Center and Diabetes Research Center Grants (P30 DK020579), the Foundation for Barnes-Jewish Hospital and the Wylie Scholar Award from the Vascular Cures Foundation.

## Author contributions

I.S. and B.R. designed the studies and wrote the manuscript. I.S., T.D.E., X.Z., S.B., C.J.S., E.S., S.A., B.D., K.B.H., P.S.M. and J.R.C. performed and analysed the experiments. I.S., T.D.E., X.Z. and S.B. prepared the figures. T.D.E., A.J., A.B., J.D.S., S.E. C.C.W., A.D., D.F. and M.A.Z. provided reagents, advised on experimental design and performed critical reading of the manuscript.

## Additional information

**Competing interests:** The authors declare no competing financial interests.

