## [Peer Review File · Nature Communications]

Reviewers' comments:

Reviewer #1 (expert in atherosclerosis)

Remarks to the Author:

This study by Sergin et al. examined the effects of macrophage TFEB overexpression and nutraceutical treatment of trehalose on macrophage autophagy-lysosomal biogenesis and progression of atherosclerosis. This senior author has previously identified transcription factor TFEB as a master regulator of autophagy-lysosomal biogenesis and the role of macrophage autophagy dysfunction in atherosclerosis. The current study further elaborated the atheroprotective potential of stimulating the autophagy-lysosome system and proposes that the natural sugar trehalose is a viable and practical method of enhancing autophagy-lysosomal biogenesis to reduce atherosclerosis. Overall, the study was well designed. Although most data were semi-quantitative immunohistochemical phenotyping, overall the data was quite compelling. The nutraceutical studies raise some questions that reduce their impact and may warrant less definitive interpretation.

Concerns:

1. The data on IL-1b is very consistent – and suggests an uncoupling of IL-1b modulation from atherosclerosis.
2. The mechanism is suggested to be via autophagy (and not IL-1/inflammasome) – but lipid metabolism, cholesterol efflux, and lysozymal function are not interrogated. It would be more complete to show such data even if negative.
3. Does macrophage-specific TFEB overexpression have effects on liver and adipose macrophages?
4. Serum trehalose following i.p. injection was mostly cleared within two hours. Therefore, the treatment strategy of “3 times of injection per week for 8 weeks” as described may not maintain a stable circulating levels of trehalose. How is this pharmacokinetic profile consistent with effect on phenotype?
5. Were lysosomal-autophagy biogenesis effects by trehalose abolished in TFEB KO macrophages?
6. There are several additional issues for the nutraceutical studies.
 - provide evidence of trehalose uptake into atherosclerotic plaque macrophages
 - what is the purity of the preparation?
 - what is the environmental and typical dietary exposure to trehalose?
 - what is the metabolism and clearance of trehalose after IP administration?
 - does IP trehalose get into cells and how?

Minor comments

1. Were all the mice on C57BL6J or C57BL6N background?
2. In Figure 4b; Supplemental Figure 3a, 3b, should the data be interpreted as “macrophage-specific TFEB-TG had no effects on serum cholesterol and other lipid parameters” instead of “Western diet feeding had no effects on serum cholesterol and other common metabolites in m^oTFEB-TG mice” as stated on Page 9?
3. The following statement might be rephrased – its not clear why this conclusion is drawn based on

the data.

"We first show that the autophagic marker LC3 and the autophagic cargo marker p62/SQSTM1 (hereafter referred to as p62) are curiously dissociated in plaque macrophages with progressive atherosclerosis. This indicates that there are subpopulations of macrophages in atherosclerotic plaques are either sufficient or deficient in mediating the autophagy process".

4. Please support the statement or rephrase "Macrophages of the atherosclerotic plaque are known to be the predominant expressors of two commonly used autophagy markers"
5. The serum concentration of trehalose was measured after 3g/kg i.p. injection but the treatment strategy in in vivo experiment used 2g/kg i.p.. Was there a specific rationale that different dosage was used?
6. Given the translational therapeutic perspective, the authors should comment on the potential for off target and adverse effects of broad activation of autophagy and lysosomal genes and functions.
7. There are several typos and grammatical errors.

Reviewer #2 (expert in atherosclerosis and immunology)

Remarks to the Author:

In this paper, the authors propose to test the feasibility of a therapeutic approach against atherosclerosis by to inducing autophagy and lysosomal function in macrophages by either TFAB overexpression or trehalose administration.

Overall comments:

The authors follow previous reports that imply TFAB in lysosomal function and autophagy and exploit this to develop a therapeutic strategy. The experimental data support the concept, however, the effects on development of atherosclerotic lesion development is rather modest when TFAB is overexpressed in macrophages. The same holds true for the results obtained with trehalose treatment; it is questionable if the modest effects on lesion development justify a therapeutic approach.

The study relies heavily on quantitative immunofluorescent stainings to study mechanisms of autophagy. Several interpretations of presented results regarding autophagy mechanisms in atherosclerotic lesions are correlative in nature and it is difficult to conclude whether the observed effects are due to overexpression of TFAB or due to secondary effects (e.g. Fig. 3). Similar caution should be exercised interpreting co-localisation results based on immunostaining (Fig. 3F).

In Fig. 4, the most impressive effect is on serum Il-1b levels, indicating a systemic anti-inflammatory effect of TFAB overexpression in macrophages. However, differences on lesion size and composition are less convincing – although the rescue experiments using TFAB TG mice crossed with ATG5 KO mice or p62 KO mice indicate that the effect of TFAB overexpression is indeed on autophagy. Further characterization of macrophage function both in vitro and in vivo in these rather complex double KO models is required to strengthen the conclusions. How were levels and functionality of macrophages affected?

Trehalose is cleared after 2 hours – so it is curious how effects can be sustained over the course of lesion development?

Reviewer #3 (expert in autophagy)

Remarks to the Author:

The link between defective autophagy in macrophages and atherosclerotic progression in both mice and humans, though the mechanisms that underlie this link are unknown. One important observation in atherosclerotic plaque macrophages is the progressive loss of autophagy and the role this loss has in lysosomal function. Specifically, the inability of autophagy-deficient, atherosclerotic macrophages to properly degrade lipids and cholesterol esters seems to be critical to this pathology. Here, the authors use the transgenic TFEB model to demonstrate that stimulating autophagic and lysosomal biogenesis can overcome genetic predisposition to atherosclerosis (ApoE model). In addition, the authors demonstrate that trehalose, a natural stabilizing agent that has been shown to induce autophagy, can both induce TFEB, autophagy, and provide atheroprotective effect in vivo in a manner dependent on the autophagy machinery.

Overall, I think this is a remarkable paper with well controlled studies that answer some long standing questions. While I admire the amount of work, the trehalose data is less compelling and should be tightened up for clarity.

1. The authors do an admirable job of demonstrating that the progressive loss of autophagy - lysosomal processes in atherosclerotic lesions. While the co-localization of p62 and LC3-II is compelling, the authors should also quantify these markers individually.
2. The authors should also demonstrate TFEB-Tg cells also contain increased levels of cleaved LC3-II via Western blot. In addition, flow analysis for LC3-II would be helpful. Also what is the level of TFEB in these transgenic mice/cells?
3. Do TFEB-Tg cells also demonstrate LAMP1 - LC3 colocalization?
4. In Figure 4k, you show LPS+OxLDL induces less than 6 pg/ml of IL1b. That is not biologically relevant and probably not significant compared to untreated cells. What level of IL1b is produced by your macrophages under basal conditions?
5. Your Figure 5e and 6e results are really fascinating, but again I would like to see the untreated controls. It's also very interesting that your LPS+OxLDL and LPS+ATP are MUCH higher than your Figure 4 values...
6. Are you able to detect trehalose in any organs after 120 minutes? The heart specifically?
7. The images in Figure 7d are not that compelling - what does 12 or 24h post trehalose look like? It seems as though that may be when your peak transcription is occurring. Also, your cells have a lot of LC3-II basally.
8. While I applaud the use of 2 antibodies, perhaps one should be in supplemental?
9. Oral dosing information should be in supplemental. Did you do an i.p. only experiment?
10. Do TFEB mice have less foam cells? Immunological characterization of the lesion should be done.
11. Recent studies have linked sugar and carbohydrates to heart disease. Did you see any effect with sucrose in terms of serum readouts?
12. Are lipids being sequestered by autophagy and p62 in foam cells? I think that a more in depth characterization of the molecular mechanisms in these foam cells and how that translates functionally. Do you get less recruitment of macrophage or are they less active or are they less (for lack of a better word) "foamy"?

Point-by-point response to Reviewer's Comments

Reviewer #1:

This study by Sergin et al. examined the effects of macrophage TFEB overexpression and nutraceutical treatment of trehalose on macrophage autophagy-lysosomal biogenesis and progression of atherosclerosis. This senior author has previously identified transcription factor TFEB as a master regulator of autophagy-lysosomal biogenesis and the role of macrophage autophagy dysfunction in atherosclerosis. The current study further elaborated the atheroprotective potential of stimulating the autophagy-lysosome system and proposes that the natural sugar trehalose is a viable and practical method of enhancing autophagy-lysosomal biogenesis to reduce atherosclerosis. Overall, the study was well designed. Although most data were semi-quantitative immunohistochemical phenotyping, overall the data was quite compelling. The nutraceutical studies raise some questions that reduce their impact and may warrant less definitive interpretation.

We thank Reviewer #1 for the comments and suggestions. We have attempted to directly address the concerns in a series of new experiments and our point by point response is below.

Concerns:

1. The data on IL-1b is very consistent – and suggests an uncoupling of IL-1b modulation from atherosclerosis.
2. The mechanism is suggested to be via autophagy (and not IL-1/inflammasome) – but lipid metabolism, cholesterol efflux, and lysozymal function are not interrogated. It would be more complete to show such data even if negative.

We agree with the reviewer and indeed, this is an interesting observation made during this work (i.e. IL-1b is reduced in the TFEB-overexpressing and Trehalose-treated models even in the absence of either autophagy and p62) leading us to conclude that IL-1beta is not a mechanistic contributor to the TFEB- and Trehalose-mediated effects on macrophages and the reduction in atherosclerosis. We particularly tested IL-1b in our models as IL-1b and the inflammasome have for a long time been one of the proposed mechanisms for atherosclerosis progression in the autophagy-deficient state. One of the conclusions for this paper is that our data points more toward a defect in autophagy/p62-dependent autophagy and the ensuing cytotoxicity rather than a reduction in IL-1b and we have noted this observation in the results and discussion sections. We would point out that there are conflicting reports in the literature regarding the role of IL-1b in atherogenesis. For example, work from Gary Owens' lab has even suggested that inactivation of IL-1 signaling in mice leads to increased atherosclerosis and features of plaque complexity (Alexander, et al. JCI 2012), work from the late Jurg Tschopp's lab shows no changes in plaque size and complexity in various NLRP3 inflammasome KO mice (Menu, et al Cell Death Disease 2011), while work from Manfred Kopf's lab has proposed a primary role for IL-1a and not the inflammasome or IL-1b in atherogenesis (Freigang, et al Nature Immunology 2014). In

future work, we still remain very interested in the underlying mechanisms for IL-1b decrease by both TFEB and trehalose but our data suggests that this decrease is not mediated by autophagy or p62 and it is unlikely to be the primary contributor to the observed atheroprotection.

With regard to point#2 (lipid metabolism and cholesterol efflux), the reviewer is astutely referring to a missing mechanism that may potentially explain some of our observations. We did not focus on this aspect rigorously due to our observations which are already published (Emanuel et al ATVB, 2014). In those studies we observed TFEB-TG macrophages have higher cholesterol efflux rates at longer time points after ApoA1 incubation (12 to 24 hours). We should note however, that these observations are again independent of autophagy as we still observed cholesterol efflux TFEB-TG macrophages on a background of autophagy deficiency (ATG5-KO) (Emanuel et al ATVB 2014). This interesting result suggests that TFEB does not need intact autophagy pathway to be able to induce cholesterol efflux. Since our ATG5 KO/TFEB-TG mice model did not show a difference in atherosclerosis, we again rule out the effect of TFEB on cholesterol efflux as a primary mechanism.

Since we had not evaluated lipid uptake and foam cell formation in TFEB-TG macrophages, we have now performed several new experiments and added them to the manuscript. First, we loaded macrophages with either DiI-oxLDL or DiI-acLDL and checked lipid accumulation in the cells by confocal microscopy. TFEB-TG macrophages did not show any difference (New Figure 4l and New Supplemental Figure 4k). We also incubated cells with oxidized LDL and acetylated LDL, extracted lipids, and quantified triglyceride levels. Again, TFEB-TG peritoneal macrophages did not show a difference in triglyceride levels after both treatments (New Supplemental Figure 4i and 4j). These experiments suggest that TFEB does not affect lipid accumulation and foam cell formation in macrophages. Finally, since lysosomal acid lipase (LAL) is a known target of TFEB, we do find increased LAL activity in TFEB-TG macrophages (New Supplemental Figure 4l) however this clearly does not affect lipid loading. To give a better explanation to the reader, we made appropriate changes in the text which points out these findings and the role of TFEB in cholesterol efflux, lipid loading, and LAL function.

3. Does macrophage-specific TFEB overexpression have effects on liver and adipose macrophages?

[UNPUBLISHED DATA REDACTED BY EDITORIAL TEAM AS PER AUTHOR REQUEST]

4. Serum trehalose following i.p. injection was mostly cleared within two hours. Therefore, the treatment strategy of “3 times of injection per week for 8 weeks” as described may not maintain a stable circulating levels of trehalose. How is this pharmacokinetic profile consistent with effect on phenotype?

This is an important question and we will discuss it in two main points: (1) precedence and practicality of performing the experiments -and- (2) an explanation of the observed biological effect.

(1) Frequent i.p. injections are rather stressful for mice. While we are performing

the injections, we also need to strike a balance with keeping the animals healthy with minimal stress as possible. In our experience, 3 times per week i.p. injections are much better tolerated than daily in the context of a 2-month experiment. We should note that this dosing frequency for trehalose has also been adapted in the majority of previously published trehalose papers with regard to neurodegeneration, etc as we have cited throughout the manuscript.

(2) Despite the practical dosing needs, the reviewer is absolutely right about questioning possible prolonged effect of trehalose due to its clearance from serum. We believe that some of this rapid clearance is actually due to trehalose uptake by tissues and we provide evidence for this in (New Figure 7b, c; New Supplemental Figure 7a). This data suggests a wider clearance window in tissues particularly for tissues such as the spleen (which contribute to the reservoir of immune cells that subsequently migrate to the atherosclerotic plaque) (New Figure 7b). Beyond that, our data also suggest that exposure to trehalose has shorter-term transcriptional effects on autophagy and lysosomal genes which results in more sustained expression levels. First, Figure 7g, h show trehalose-mediated increases in

transcriptional activation preceding the rise in protein levels (i.e. suggesting that once the transcriptional program has initiated, the trehalose effect would be there even trehalose itself is totally cleared). But to more directly support this conclusion we performed a new experiment in which we incubate macrophages with trehalose for 3 hours and then washed off the trehalose, changing to regular medium. After waiting for another 9 hours, the levels of two critical autophagy markers (p62 and LC3) still remain elevated (New Supplemental Figure 7k). Overall, we believe that despite the clearance of extracellular (or serum) trehalose, its intracellular effect on autophagy can clearly last longer supporting more sustained boosts in autophagy on days in which trehalose is administered in vivo.

5. Were lysosomal-autophagy biogenesis effects by trehalose abolished in TFEB KO macrophages?

This is an excellent question and it is one of the current focuses of our lab. We think that TFEB is not the only link through which trehalose provides its transcriptional effects. TFEB is a member of a transcription factor family consisting of mTF, TFEB, and TFE3. It has been proposed that these transcription factors could have overlapping roles in different cell types/tissues. Thus, we also tested how trehalose affects the other two transcription factors and have added this data to the manuscript (New Figure 8c-e; New Supplemental Figure 8b-e). Based on our immunoblotting and immunofluorescence studies, trehalose has a similar activating effect on TFE3 (i.e. trehalose induces TFE3 expression and its nuclear localization). This was not observed for mTF (interestingly, mTF nuclear localization was quite high at baseline and did not increase with trehalose treatment). Based on these observations, we do not expect a total reversal of trehalose autophagy induction in TFEB-KO cells but a partial decrease could be observed. We are in the process of creating and gathering the required tools for those studies at present (i.e. TFEB-flox mice, TFE3-KO mice, TFE3-TFEB double KO, etc.) but these are currently longer-term propositions which also require initial characterization of the TFEB, TFE3, and double KO mice and the respective macrophages rendering this effort beyond the scope of the current manuscript.

6. There are several additional issues for the nutraceutical studies.
-provide evidence of trehalose uptake into atherosclerotic plaque macrophages

Although the reviewer has a valid point, the low cellular yield from such an effort make it extremely difficult to actually observe trehalose in plaque macrophages. Generally, obtaining plaque macrophages from the aorta is challenging and limits many investigators' ability to perform desired experiments. Despite this, we did attempt to detect trehalose by mass spectrometry from macrophages isolated from pooled atherosclerotic aortas but the amount of cells we could isolate from these samples were never sufficient for such analysis.

Since trehalose kinetics and uptake is an issue raised by all reviewers, we would like to provide a summary of our approach for evaluating trehalose uptake into tissues and cells. First, we would like to mention a few important background points:

- 1- An accurate evaluation of trehalose uptake can only be considered in the context of trehalose clearance and one of the main mechanisms of trehalose clearance is tissue

trehalase (the enzyme that cleaves the disaccharide into 2 glucose molecules). The kidneys, liver, and GI tract are known to be high expressors of trehalase but all cells appear to have some level of expression. In this regard, an extensive study of trehalase levels and activity in different tissues will be needed to get an accurate assessment of trehalase kinetics.

- 2- There is quite a large trehalose literature already published with several reports proposing various mechanisms for trehalose cellular uptake. One proposed mechanism is uptake via fluid phase endocytosis (Wolkers et al *Cryobiology* 2001 and Wolkers et al *Biochim Biophys Acta* 2003). This is quite intriguing since that would suggest eventual trehalose localization to the endosome/lysosome system which could link trehalose to a compartment mechanistically linked to autophagy-lysosomal biogenesis. A recent study has suggested that trehalose can also be taken up by carrier-mediated transport (Mayer *et al. Scientific Reports*, 2016).

Although a study of the kinetics/clearance of trehalose and mechanisms of uptake into tissues/cells will in general require extensive and detailed studies, we have now performed several *in vivo* and *in vitro* experiments to hopefully provide a satisfactory first-pass analysis of this subject.

In *in vivo* experiments, we administered trehalose to mice, sacrificed them at different time points and checked trehalose levels in several tissues (aorta, heart, spleen) by mass spectrometry analysis. We chose these tissues due to their relevance to our disease model (aorta and heart) or presence of large immune cell reservoir (spleen). We also confirmed our aorta results with an independent method (a colorimetric trehalose assay). To summarize, tissue trehalose levels showed somewhat similar trend that we observed in serum. We detected highest level after one hour and levels started to decrease at two hours but it was still higher than baseline after four hours. There was a difference between the trehalose kinetics in different tissues with spleen showing a persistence of trehalose even after four hours of *i.p.* injection (New Figure 7b). We were also able to recapitulate our aorta mass spectrometry results via the colorimetric method (New Supplemental Figure 7a). To evaluate trehalose cellular uptake, we treated macrophages with trehalose and again detected trehalose levels in cell lysates by mass spectrometry (New Figure 7c). This *in vitro* assay again supports the premise that trehalose can be taken up by macrophages within the time and concentration levels of our *in vivo* measures of trehalose kinetics.

-what is the purity of the preparation?

We used D-(+)-Trehalose dihydrate – from *Saccharomyces cerevisiae* purchased from Sigma (T0167) with a purity >99% based on HPLC. It has also passed cell culture testing by the company. After dissolving the powder, we again filter-sterilize the solution through a 0.22 μm filter system and aliquot under sterile conditions for subsequent cell culture work or administration to mice.

-what is the environmental and typical dietary exposure to trehalose?

We would recommend this review as a good general source for trehalose: Richards AB, *et al.* Trehalose: a review of properties, history of use and human tolerance, and results of multiple safety studies. *Food and Chemical Toxicology*. 2002.

Based on this reference various organisms contain α,α -trehalose including plants, algae, fungi, yeasts, bacteria, insects and invertebrates. For example, it is the principal sugar (approximately 80–90%) for all examined insects (found in the hemolymph) and can constitute about 20% of all carbohydrates during specific stages of insect development. Of note, trehalose is not found naturally in higher species like mammals. Human consumption is also summarized in the same reference: “Modern food sources may contain substantial quantities of trehalose. Some of these include honey (0.1–1.9%), mirin (1.3–2.2%), sherries (<10–391 mg/l), brewer’s (0.01–5.0%) and baker’s yeasts (15–20%), and therefore most items made using yeast. Commercially grown mushrooms can contain 8–17% (w/w) trehalose. It also occurs in invertebrates such as lobster (2.5 mg/100 ml blood), crab (1.5 mg/100 ml blood) and prawns (0.5% dw). Trehalose has also been isolated from various seed plants, including sunflowers.” Interestingly, it has been proposed that although the level of trehalose is not currently a significant part of the modern human diet, it was a relatively large dietary component of ancient man.

-what is the metabolism and clearance of trehalose after i.p. administration?

As we discussed above, tissue trehalase is a major mechanism by which trehalose is metabolized (see also Richards AB et al Food and Chemical Toxicology 2002). Trehalase is an enzyme present at high levels in the kidneys, GI tract, and liver but is detectable in many tissues. It is responsible for cleaving trehalose to two glucose molecules. One of the reasons we focused on i.p. administration is to bypass the GI tract/Liver degradation. In fact, this is likely the primary reason why oral trehalose is not atheroprotective (Figure 9j). As mentioned above, an extensive study of trehalase levels and activity in different tissues will be needed to get an accurate assessment of trehalose kinetics.

-does i.p. trehalose get into cells and how?

Please see above for the detailed answer to this question. We have provided several new experiments on trehalose tissue kinetics and uptake into relevant tissues and macrophages by mass spectrometry and colorimetric assays (New Figure 7b, c; New Supplemental Figure 7a). As we also mentioned above, different studies have suggested different uptake mechanisms (for example Wolkers *et al.* Cryobiology, 2001; Wolkers et al. Biochim Biophys Acta, 2003; Mayer *et al.* Scientific Reports, 2016). The possibility of fluid phase endocytosis is an interesting one since it suggests eventual trehalose localization to the endosome/lysosome system which could link trehalose to a compartment mechanistically linked to autophagy-lysosomal biogenesis. The trehalose uptake kinetics and mechanisms again requires extensive and detailed studies which are beyond the scope of this manuscript.

Minor comments

1. Were all the mice on C57BL6J or C57BL6N background?

Mice are on C57BL/6J background. This information is added to the appropriate place in experimental procedures.

2. In Figure 4b; Supplemental Figure 3a, 3b, should the data be interpreted as “macrophage-specific TFEB-TG had no effects on serum cholesterol and other lipid parameters” instead of “Western diet feeding had no effects on serum cholesterol and other common metabolites in mTFEB-TG mice” as stated on Page 9?

We thank to the reviewer for the suggested correction and this sentence has been corrected.

3. The following statement might be rephrased – its not clear why this conclusion is drawn based on the data.

“We first show that the autophagic marker LC3 and the autophagic cargo marker p62/SQSTM1 (hereafter referred to as p62) are curiously dissociated in plaque macrophages with progressive atherosclerosis. This indicates that there are subpopulations of macrophages in atherosclerotic plaques are either sufficient or deficient in mediating the autophagy process”.

We thank to the reviewer for the suggested correction. This sentence has been revised.

4. Please support the statement or rephrase “Macrophages of the atherosclerotic plaque are known to be the predominant expressors of two commonly used autophagy markers”

This statement is based on our evaluation of several autophagy markers including LC3 and p62 in atherosclerotic plaques in previously published work which has been cited in the references (Razani et al Cell Metab 2012; Sergin et al. Science Signaling 2016).

5. The serum concentration of trehalose was measured after 3g/kg i.p. injection but the treatment strategy in in vivo experiment used 2g/kg i.p.. Was there a specific rationale that different dosage was used?

In our first series of experiments, our goal was to give an equivalent maximum dose of trehalose we could via either i.p. and oral gavage routes – The concentration of trehalose in solution capable of being administered in this comparison turns out to be 3 g/kg (Figure 7a and 9i). Having seen that this dosing i.p. can achieve as high as 10 mM trehalose in the serum, we reduced our dosing for the longer term studies to match previous published studies of trehalose in the neurodegenerative and other arenas as we have cited.

6. Given the translational therapeutic perspective, the authors should comment on the potential for off target and adverse effects of broad activation of autophagy and lysosomal genes and functions.

We have now added a section regarding this point in the discussion section. In particular, one of the concerns in the autophagy field is the possibility for hyperactive autophagy to induced autophagy-mediated cell death (also called autosis) leading to deleterious outcomes. The dose of trehalose and its downstream effects do not appear to bear this out. In fact, our data with trehalose shows a protection from lipid-induced macrophage death

and reduced cell death in the plaque.

7. There are several typos and grammatical errors.

Thank you for the comment - we have reevaluated the manuscript and hopefully have made all the appropriate changes.

Reviewer #2:

In this paper, the authors propose to test the feasibility of a therapeutic approach against atherosclerosis by inducing autophagy and lysosomal function in macrophages by either TFAB overexpression or trehalose administration.

We thank Reviewer #2 for the comments and suggestions. We have attempted to directly address the concerns in a series of new experiments and our point by point response is below.

Overall comments:

The authors follow previous reports that imply TFAB in lysosomal function and autophagy and exploit this to develop a therapeutic strategy. The experimental data support the concept, however, the effects on development of atherosclerotic lesion development is rather modest when TFAB is overexpressed in macrophages. The same holds true for the results obtained with trehalose treatment, it is questionable if the modest effects on lesion development justify a therapeutic approach.

We agree that the degree of effect on lesion size is always an important parameter in gauging the potential therapeutic impact of a newly proposed pharmacologic. There are other critical parameters as well such as effects on immune cell content and features of plaque complexity including degree of apoptosis and necrotic core formation. Plaque complexity is often used as a surrogate of plaque stability with clear therapeutic implications. It should be noted that in these studies we have attempted to evaluate not just lesion size but these other parameters as well and have found that measures of plaque complexity are reduced even further by TFEB-overexpression while again being completely abrogated in the dual ATG5-KO and p62-KO settings (e.g. Figure 4e,f vs New Figure 5b,c vs New Figure 6b,c).

Overall, hopefully the reviewer would agree with us that any studies of atherosclerosis using mouse models are largely proof-of-principle concepts that should stimulate further translational research (they are clearly not stand-alone in determining the actual therapeutic impact). This is what we desire as an outcome of this paper: to stimulate the community to further examine these autophagy-lysosomal biogenesis pathways in translational cardiovascular studies. We should also note that this study is one of the first to actually demonstrate an atheroprotective benefit to stimulating macrophage autophagy. The role of macrophage autophagy in atherosclerosis was directly studied in several papers

in 2012 (Razani et al Cell Metab 2012 and Liao et al Cell Metab 2012). In the past decade, the only well-studied autophagy-modulating pharmacologic that has been demonstrated to be atheroprotective is rapamycin (the classic mTOR inhibitor and thus autophagy stimulator). However, the specificity of rapamycin to autophagy is questionable and the clinical utility of rapamycin is fraught with side effects. Thus, we believe these studies on TFEB and trehalose should be a welcome addition to the field.

The study relies heavily on quantitative immunofluorescent stainings to study mechanisms of autophagy. Several interpretations of presented results regarding autophagy mechanisms in atherosclerotic lesions are correlative in nature and it is difficult to conclude whether the observed effects are due to overexpression of TFEB or due to secondary effects (e.g. Fig. 3). Similar caution should be exercised interpreting co-localisation results based on immunostaining (Fig. 3F).

We understand the reservations of the reviewer, however we should point out the evaluation of autophagy is ordinarily very difficult since the process is very dynamic in vivo and in cell culture and presently available markers and assays have known limitations in detecting this dynamic nature (i.e. autophagic flux). We believe we have conducted a wide-ranging set of experiments using the recommended strategies used in the field (much of which requires stainings and quantitative immunofluorescence). This involves utilizing both LC3 and p62 markers and GFP-LC3 mice. Many autophagy experts we have learned from over the years utilize these approaches. We would also like to point out:

- 1- We have also now conducted FACS analysis on atherosclerotic aortas to compare plaque macrophages from control and TFEB-TG mice and in agreement with the other data presented, we find increased co-expression of p62, LC3, and Lamp2 in TFEB-TG plaque macrophages (New Figure 3h, i).**
- 2- The cumulative findings in Figure 3 and 4 (on TFEB-overexpression) should be considered together with Figures 5, 6, and all related supplemental data where essentially all the beneficial features of TFEB-overexpression are abrogated in either the absence of the essential autophagy marker ATG5 or the selective autophagy chaperone p62. Based on this, we conclude that that TFEB's salutary effects are autophagy/selective autophagy dependent.**

In Fig. 4, the most impressive effect is on serum Il-1b levels, indicating a systemic anti-inflammatory effect of TFEB overexpression in macrophages. However, differences on lesion size and composition are less convincing – although the rescue experiments using TFEB TG mice crossed with ATG5 KO mice or p62 KO mice indicate that the effect of TFEB overexpression is indeed on autophagy. Further characterization of macrophage function both in vitro and in vivo in these rather complex double KO models is required to strengthen the conclusions. How were levels and functionality of macrophages affected?

We performed several new experiments taking into account the reviewer's suggestions:

- 1- We have added several new analyses to the atherosclerotic plaque phenotypes of the double transgenic/KO cohorts (i.e. the macrophage TFEB-TG/ATG5-KO and macrophage TFEB-TG/p62-KO) in order to directly compare various features of plaque complexity with macrophage TFEB-TG mice. In contrast to the TFEB-TG**

plaques which showed decreased macrophage content and apoptotic cell death and necrotic cores, the double KO groups displayed no such (New Figure 5b, c; New Figure 6b, c). We did also attempt to measure serum IL-1b levels from the double KO cohorts but the repeated use of these samples (including freeze-thawing) for metabolite analysis had left very little usable serum for such analysis.

- 2- In order to demonstrate that dual TFEB-TG/ATG5-KO macrophages still maintain a complete absence of autophagy or autophagic progression, we have added New Supplemental Figure 5a showing the absence of LC3 II conversion (indicating continued absence of autophagy) in cultured macrophages.
- 3- As also we discussed above (in the comments to Reviewer #1), we have previously conducted several assays on the lipid handling of TFEB-TG macrophages (Emanuel et al ATVB, 2014). In those studies we observed TFEB-TG macrophages have higher cholesterol efflux rates at longer time points after ApoA1 incubation (12 to 24 hours). However, akin to the IL-1b data, these effects were again independent of autophagy as we still observed cholesterol efflux TFEB-TG macrophages on a background of autophagy deficiency (ATG5-KO) (Emanuel et al ATVB 2014). These results add to the IL-1b data by suggesting that TFEB does not need an intact autophagy pathway to either suppress IL-1b levels or to induce cholesterol efflux. Since our ATG5 KO/TFEB-TG mice model did not show a difference in atherosclerosis, we rule out the effect of TFEB on IL-1b or cholesterol efflux as primary mechanisms by which plaque burden and complexity are reduced. We instead favor a reduction in apoptosis mediated by removal of p62-enriched polyubiquitinated protein aggregates as a more likely mechanism. This conclusion is buttressed by the TFEB-TG/p62-KO data (Figure 6 including the newly added components).
- 4- In order to more fully characterize the lipid handling of TFEB-TG cells (aside from cholesterol efflux that we had assessed in Emanuel et al ATVB, 2014), we also performed several new experiments in this regard. First, we loaded macrophages with either DiI-oxLDL or DiI-acLDL and checked lipid accumulation in the cells by confocal microscopy. TFEB-TG macrophages did not show any difference (New Figure 4l and New Supplemental Figure 4k). We also incubated cells with oxidized LDL and acetylated LDL, extracted lipids, and quantified triglyceride levels. Again, TFEB-TG peritoneal macrophages did not show a difference in triglyceride levels after both treatments (New Supplemental Figure 4i and 4j). These experiments suggest that TFEB does not affect lipid accumulation and foam cell formation in macrophages. Based on this data we again conclude that foam cell formation and altered lipid handling is not a major reason for the atheroprotective phenotype of macrophage TFEB-TG mice.

Trehalose is cleared after 2 hours – so it is curious how effects can be sustained over the course of lesion development?

Although a study of the kinetics/clearance of trehalose and mechanisms of uptake into tissues/cells will in general require extensive and detailed studies, we have now performed several in vivo and in vitro experiments to hopefully provide a satisfactory first-pass analysis of this subject.

We believe that some of this clearance from the serum is actually due to trehalose uptake by tissues and we provide evidence for this in (New Figure 7b, c; New Supplemental Figure 7a). These data suggest a wider clearance window in tissues particularly for tissues such as the spleen (which contribute to the reservoir of immune cells that subsequently migrate to the atherosclerotic plaque) (New Figure 7b). Beyond that, our data also suggest that exposure to trehalose has shorter-term transcriptional effects on autophagy and lysosomal genes which results in more sustained expression levels. First, Figure 7g, h show trehalose-mediated increases in transcriptional activation preceding the rise in protein levels (i.e. suggesting that once the transcriptional program has initiated, the trehalose effect would be there even trehalose itself is totally cleared). But to more directly support this conclusion we performed a new experiment in which we incubate macrophages with trehalose for 3 hours and then washed off the trehalose, changing to regular medium. After waiting for another 9 hours, the levels of two critical autophagy markers (p62 and LC3) still remain elevated (New Supplemental Figure 7k). Overall, we believe that despite the clearance of extracellular (or serum) trehalose, its intracellular effect on autophagy can clearly last longer supporting more sustained boosts in autophagy on days in which trehalose is administered in vivo. If such stimulation is being provided 3x/week, it is conceivable that atherosclerosis can be reduced (akin to why neurodegenerative diseases and others can be ameliorated by 3x/week trehalose administration as has been reported before and discussed in the manuscript).

Reviewer #3:

The link between defective autophagy in macrophages and atherosclerotic progression in both mice and humans, though the mechanisms that underlie this link are unknown. One important observation in atherosclerotic plaque macrophages is the progressive loss of autophagy and the role this loss has in lysosomal function. Specifically, the inability of autophagy-deficient, atherosclerotic macrophages to properly degrade lipids and cholesterol esters seems to be critical to this pathology. Here, the authors use the transgenic TFEB model to demonstrate that stimulating autophagic and lysosomal biogenesis can overcome genetic predisposition to atherosclerosis (ApoE model). In addition, the authors demonstrate that trehalose, a natural stabilizing agent that has been shown to induce autophagy, can both induce TFEB, autophagy, and provide atheroprotective effect in vivo in a manner dependent on the autophagy machinery.

Overall, I think this is a remarkable paper with well controlled studies that answer some long standing questions. While I admire the amount of work, the trehalose data is less compelling and should be tightened up for clarity.

We also thank Reviewer #3 for the comments and suggestions and our point by point response is below.

1. The authors do an admirable job of demonstrating that the progressive loss of autophagy-lysosomal processes in atherosclerotic lesions. While the co-localization of p62 and LC3-II is compelling, the authors should also quantify these markers individually.

Several of the series of images taken from atherosclerotic plaques have been already quantified (such as comparison of early vs advanced lesion in mice, Figure 1B, 1C; and TFEB-TG vs control lesions, Figure 3E). The quantitation was indeed missing for the human atherosclerotic samples and we have now performed the suggested quantification (New Supplemental Figure 1C and 1D). Similar to the mouse atherosclerosis quantification, LC3 levels were significantly lower in maximally diseased regions of the human plaque samples. We also detected an accumulation for p62 in maximally disease regions although it did not reach statistical significance.

2. The authors should also demonstrate TFEB-Tg cells also contain increased levels of cleaved LC3-II via Western blot. In addition, flow analysis for LC3-II would be helpful. Also what is the level of TFEB in these transgenic mice/cells?

We agree with the reviewer about better characterization of autophagy in TFEB-TG macrophages. As suggested we checked the p62 and LC3 levels in TFEB-TG peritoneal macrophages either in basal condition or after 1 and 3 hours of bafilomycin treatment by western blotting (New Figure 2d). In agreement with high mRNA transcript levels of p62 (Figure 2c), p62 levels were always higher in TFEB-TG cells. Also as expected LC3 I levels was higher in TFEB-TG macrophages at baseline and after bafilomycin treatment this increase was reflected in a shift to LC3 II band. Also as suggested we analyzed atherosclerotic aorta macrophages in TFEB-TG mice by FACS analysis and found 1) increases in LC3 and Lamp1 staining and 2) in agreement with our plaque immunofluorescence stainings, TFEB-TG plaque macrophage populations expressing higher levels of LC3 & p62 and LC3 & Lamp2 (New Figure 3h and 3i). Finally, we checked TFEB expression levels in TFEB-TG peritoneal macrophages finding both increased (5-8 fold) transcript (Figure 2a) and protein expression (New Supplemental Figure 2a).

3. Do TFEB-Tg cells also demonstrate LAMP1 – LC3 colocalization?

To be able to answer this question, we performed two experiments:

- 1- We conducted live cell imaging tracking the autophagosome marker (GFP-LC3) and lysosome marker (LysoTracker) in Control and TFEB-TG macrophages. Over a 10 minute-long experiment, we observed increased co-localization of GFP-LC3 and LysoTracker in TFEB-TG macrophages (New Figure 2g, New Supplemental Figure 2f-h). We should note that since TFEB induces both autophagy and lysosome biogenesis, this autophagosome-lysosome interaction could also be contributed to by the larger available pool of these subcellular structures.
- 2- We analyzed atherosclerotic plaque macrophages derived from control and macrophage TFEB-TG mice by FACS staining with LC3 and Lamp2 (Figure 3i). We noted TFEB-TG macrophages also have increased and coincident expression of LC3 and Lamp2. Although this experiment is an indirect measure of co-localization of these markers, it does support the notion that autophagosome and lysosome pools are enhanced in macrophage TFEB-TG atherosclerotic plaques in vivo.

4. In Figure 4k, you show LPS+OxLDL induces less than 6 pg/ml of IL1b. That is not biologically relevant and probably not significant compared to untreated cells. What level of IL1b is produced by your macrophages under basal conditions?

This is an astute observation by the reviewer which we would like to clarify. It is known that oxLDL is a poor inflammasome stimulator compared to the classically potent DAMPs (danger associated molecular patterns) such as crystalline material (in our case cholesterol crystals) or ATP. Regardless of the inflammasome stimuli used, the resultant IL-1b production will also be heightened in the setting of either ATG5-KO or p62-KO as these macrophages have activated inflammasome signaling (described previously by us and others – see Razani B et al Cell Metabolism 2012 and Sergin I et al Science Signaling 2016). The primary reason we presented these stimuli was to demonstrate and compare TFEB's IL-1b dampening effect under three different conditions (with notably no effect on non-inflammasome-mediated cytokines like TNF α). In order to reduce confusion, we moved the oxLDL data to the supplemental now (Supplemental Figure 4g, 5f, and 6e) and kept the more potent inflammasome inducers (cholesterol crystals and ATP) in the main figures.

As for the basal levels of IL-1b produced by macrophages, we would also like to clarify this point. For a macrophage to be able to secrete IL1b, there is a need for two signals. The first signal induces the transcriptional/translational machinery which generates the precursor protein pro-IL1b. The second signal is the cleavage of pro-IL1b to mature IL1b by the protease Caspase1 and this requires activation of the inflammasome. It is only then that mature IL1b can be secreted from the cell. In most inflammasome/IL1b assays, the first signal used LPS and the second signal is the DAMP (like the more potent cholesterol crystal or ATP or the less potent oxLDL). In a perfect system LPS-only treatment (without a DAMP) will lead to negligible IL1 β secretion. We do such controls usually to make sure our LPS is not contaminated. We have attached a representative result showing no IL1 β secretion in untreated macrophages and very minimal IL1 β secretion in LPS-only treated macrophages (see below).

5. Your Figure 5e and 6e results are really fascinating, but again I would like to see the untreated controls. It's also very interesting that your LPS+OxLDL and LPS+ATP are MUCH higher than your Figure 4 values...

Our comments here should be taken together with our comments for #4 above. It should be noted that Figure 5 and Figure 6 are evaluating the IL-1b response for TFEB in the background of ATG5-KO and p62-KO macrophages respectively. As we mentioned above, ATG5-KO and p62-KO macrophages have a hyperactive inflammasome system (as we and others have published before – see Razani B et al Cell Metabolism 2012 and Sergin I et al Science Signaling 2016). This is why the results of the IL-1b are higher than the “wildtype/C57BL6” background Figure 4 experiments. All that is being seen is the hyperactivity of IL-1b being brought out by the concomitant ATG5-KO (Figure 5) or p62-KO (Figure 6). That being said, each of these experiments are designed simply to compare the presence and absence of TFEB overexpression. The cohorts of mice used to obtain the macrophages and the yield/plating density of macrophages during each experiment dictate the exact levels of secreted IL-1b measured from experiment to experiment. Thus, we do not believe each of these data sets is precisely comparable. There will be predicted trends (such as the ATG5-KO and p62-KO cells will in general have higher IL-1b than the control cells of Figure 4 for the reasons stated above). But direct comparisons are not valid unless all cells are extracted and plated at same density in one experiment. Again, these experiments are simply designed to evaluate the effect of TFEB-overexpression in each condition or genetic background.

6. Are you able to detect trehalose in any organs after 120 minutes? The heart specifically?

Since trehalose kinetics and uptake is an issue raised by all reviewers, we would like to provide a summary of our approach for evaluating trehalose uptake into tissues and cells. First, we would like to mention a few important background points:

- 1- An accurate evaluation of trehalose uptake can only be considered in the context of trehalose clearance and one of the main mechanisms of trehalose clearance is tissue trehalase (the enzyme that cleaves the disaccharide into 2 glucose molecules). The kidneys, liver, and GI tract are known to be high expressors of trehalase but all cells appear to have some level of expression. In this regard, an extensive study of trehalase levels and activity in different tissues will be needed to get an accurate assessment of trehalose kinetics.
- 2- There is quite a large trehalose literature already published with several reports proposing various mechanisms for trehalose cellular uptake. One proposed mechanism is uptake via fluid phase endocytosis (Wolkers et al Cryobiology 2001 and Wolkers et al Biochim Biophys Acta 2003). This is quite intriguing since that would suggest eventual trehalose localization to the endosome/lysosome system which could link trehalose to a compartment mechanistically linked to autophagy-lysosomal biogenesis. A recent study has suggested that trehalose can also be uptaken by carrier-mediated transport (Mayer *et al.* Scientific Reports, 2016).

Although a study of the kinetics/clearance of trehalose and mechanisms of uptake into tissues/cells will in general require extensive and detailed studies, we have now performed several in vivo and in vitro experiments to hopefully provide a satisfactory first-pass analysis of this subject.

In in vivo experiments, we administered trehalose to mice, sacrificed them at different time points and checked trehalose levels in several tissues (aorta, heart, spleen) by mass

spectrometry analysis. We chose these tissues due to their relevance to our disease model (aorta and heart) or presence of large immune cell reservoir (spleen). We also confirmed our aorta results with an independent method (a colorimetric trehalose assay). To summarize, tissue trehalose levels showed somewhat similar trend that we observed in serum. We detected highest level after one hour and levels started to decrease at two hours but it was still higher than baseline after four hours. There was a difference between the trehalose kinetics in different tissues with spleen showing a persistence of trehalose even after four hours of i.p. injection (New Figure 7b). We were also able to recapitulate our aorta mass spectrometry results via the colorimetric method (New Supplemental Figure 7a). To evaluate trehalose cellular uptake, we treated macrophages with trehalose and again detected trehalose levels in cell lysates by mass spectrometry (New Figure 7c). This *in vitro* assay again supports the premise that trehalose can be taken up by macrophages within the time and concentration levels of our *in vivo* measures of trehalose kinetics.

7. The images in Figure 7d are not that compelling – what does 12 or 24h post trehalose look like? It seems as though that may be when your peak transcription is occurring. Also, your cells have a lot of LC3-II basally.

We have performed similar experiments for different time points (both by using GFP-LC3 and LC3 staining as readouts) and we observe that the effect of trehalose is most prominent at 6h for immunofluorescence analysis lowering by 12h (LC3 quantification shown in New Supplemental Figure 7h). That is one reason why we performed the bulk of our analyses at the 6h time point. That time point also matches our transcription data showing increase in transcripts beginning at 3 hour and in most cases peaking at 6 hours (Figure 7g). Overall, based on the cumulative data (IF, qPCR and western blot), we conclude that trehalose's maximal effect is between the 6 to 12 hour period.

With regard to LC3 levels, indeed macrophages have a high level of LC3-II basally. This is likely characteristic of this cell type as they are the predominant degradative cell of the body likely requiring elevated basal autophagy (we and others have previously corroborated this as well - Choi et al *Immunity* 2014; DeSelm et al *Dev. Cell.* 2011; Razani B et al *Cell Metabolism* 2012, etc.)

8. While I applaud the use of 2 antibodies, perhaps one should be in supplemental?

Based on the reviewer's suggestion, the 2nd antibody data is moved to supplemental data now (Now Supplemental Figure 3A, 9A, and 9B).

9. Oral dosing information should be in supplemental. Did you do an i.p. only experiment?

We believe that reviewer made that comment because we showed the effect of trehalose i.p. injection in Figure 7a and little effect is observed in oral trehalose treatment as shown in Figure 9i. However we believe that this is an important piece of data. As the reviewer is probably well aware, there is a quite a large body of trehalose literature where the *in vivo* experiments are performed with oral-only trehalose administration. It would be important for the scientific community to see head-to-head comparisons of oral-only and i.p.-dosing in primary data to realize that much of the orally dosed form does not make it into the

circulation (likely due to the effect of the enzyme trehalase). We hope such comprehensive assessment of trehalose in vivo would be noted by scientists evaluating trehalose in other disease models and to correlate the efficacy of oral dosing with serum levels.

With regard to an i.p. only experiment, we limited this evaluation to the trehalose kinetics – since we had performed all our atherosclerosis experiments with a combination of i.p./oral dosing and we maintained this throughout the manuscript in the 9 large cohorts we analyzed including the sucrose and maltose cohorts (Figures 9h, Figure 10a, e, Supplemental Figure 9f). In fact it was when we recognized that oral-dosing was not appearing in the circulation as readily at the i.p. dosing was when we conducted the oral-only trehalose atherosclerosis experiment (Figure 9j).

10. Do TFEB mice have less foam cells? Immunological characterization of the lesion should be done.

We have now performed several new experiments and added them to the manuscript. First, we loaded macrophages with either DiI-oxLDL or DiI-acLDL and checked lipid accumulation in the cells by confocal microscopy. TFEB-TG macrophages did not show any difference (New Figure 4l and New Supplemental Figure 4k). We also incubated cells with oxidized LDL and acetylated LDL, extracted lipids, and quantified triglyceride levels. Again, TFEB-TG peritoneal macrophages did not show a difference in triglyceride levels after both treatments (New Supplemental Figure 4i and 4j). These experiments suggest that TFEB does not affect lipid accumulation and foam cell formation in macrophages.

With regard to the immunological characterization, we further corroborated the macrophage content of the Control and macrophage TFEB-TG plaques by comparing the immunofluorescence staining and subsequent quantification of 2 well-known macrophage markers (MOMA-2 and CD68). In TFEB-TG mice, both stainings showed lower macrophage content in the TFEB-TG plaques (although CD68 did not reach statistical significance) (Figure 4e, New Supplemental Figures 4c-e). The same type of comparison in the TFEB-TG cohorts (in the background of macrophage-specific ATG5-KO and p62-KO backgrounds) revealed the reduction of macrophage content had been abrogated (New Figure 5b and 6b).

11. Recent studies have linked sugar and carbohydrates to heart disease. Did you see any effect with sucrose in terms of serum readouts?

We performed serum metabolite analysis for sucrose-treated groups and did not detect any metabolite difference in those groups compared to controls. New data have been added to the appropriate figures (Figure 9g, Supplemental Figures 9d, e)

12. Are lipids being sequestered by autophagy and p62 in foam cells? I think that a more in depth characterization of the molecular mechanisms in these foam cells and how that translates functionally. Do you get less recruitment of macrophage or are they less active or are they less (for lack of a better word) “foamy”?

This comment is somewhat related to comment #10 where we described our approach and new experiments evaluating foam cell formation. However, the reviewer's idea about the sequestration of the lipids is quite interesting. We had also considered this while we were trying to characterize p62-enriched inclusion bodies that form in autophagy-deficient states (Sergin et al Science Signaling 2016). Thus, we incubated macrophages with oxLDL and co-stained them with either of the lipid markers BODIPY or Nile Red together with p62 (New Supplemental Figure 4f). Our overall conclusion is that accumulated lipids are not prominently surrounded by p62 and p62 inclusion bodies/aggregates do not contain a significant amount of lipids. That being said, the concept of lipophagy in atherosclerosis and cardiometabolic disease is clearly understudied and a very important line of future investigation. These studies will clearly require multiple tools and experimental approaches for a more nuanced evaluation.

Reviewers' comments:

Reviewer #1 (Remarks to the Author):

The authors have addressed in great detail all reviewers' comments and the newly reported data are convincing. In the rebuttal letter, the authors showed the role of TFEB-Tg in adipose tissue macrophage autophagy, suggesting broader systemic effects of TFEB-Tg.

Modest concerns:

1. Provide data for macrophages from total TFEB KO mice that addresses "Were lysosomal-autophagy biogenesis effects by trehalose abolished in TFEB KO macrophages?". The data provided on trehalose effects on TFE3 are very interesting, but the question here is how much is mediated by TFEB – and the authors should be able to address this without DKO or tissue-specific KOs. How would the authors interpret a finding of similar effects on lysosomal-autophagy biogenesis in TFEB KO and WT macrophages?
2. It might be worth discussing the mechanisms of reduced macrophage content in the TFEB-Tg mice lesion, e.g. through reduced monocytopoiesis, reduced monocyte infiltration, increased clearance (efferocytosis) etc.
3. The schematic figure may also incorporate that TFEB-Tg reduced lesion macrophage content, increased macrophage cholesterol efflux (partially dependent on autophagy based on Emanuel et al. ATVB paper), suppressed IL-1 β secretion (independent of autophagy), increased lysosomal acid lipase activity, but did not affect lipid uptake and accumulation based on the new data provided.

Reviewer #2 (Remarks to the Author):

The authors did an excellent job addressing this reviewers concerns. Addition of new data strengthens the conclusions. Overall, this is an important study providing significant findings that will be helpful for many researchers in the field.

No more comments.

Reviewer #3 (Remarks to the Author):

The authors have done a very complete and remarkable job addressing all of my concerns in this manuscript. I believe that they have significantly strengthened an already important paper with their new studies.

Sergin, et al. Exploiting Macrophage Autophagy-Lysosomal Biogenesis as a Therapy for Atherosclerosis

Point-by-point response to Reviewer's Comments

Reviewer #1:

The authors have addressed in great detail all reviewers' comments and the newly reported data are convincing. In the rebuttal letter, the authors showed the role of TFEB-Tg in adipose tissue macrophage autophagy, suggesting broader systemic effects of TFEB-Tg.

We thank Reviewer #1 for these additional comments/suggestions and have attempted to address these concerns in a point by point response as follows.

1. Provide data for macrophages from total TFEB KO mice that addresses "Were lysosomal-autophagy biogenesis effects by trehalose abolished in TFEB KO macrophages?". The data provided on trehalose effects on TFE3 are very interesting, but the question here is how much is mediated by TFEB – and the authors should be able to address this without DKO or tissue-specific KOs. How would the authors interpret a finding of similar effects on lysosomal-autophagy biogenesis in TFEB KO and WT macrophages?

Amongst the MiTF/TFE transcription factor family, TFEB is particularly important for embryonic development and unfortunately TFEB-KO mice die in utero at ~E9.5-E10.5 from placental defects (Steingrimsson E et al Development 1998). Thus, TFEB-KO mice cannot be used and tissue-specific TFEB-KO mice are the only way to study the macrophage functions of this transcription factor (e.g. LysM-Cre x TFEB-flox). We are in the process of developing these mice along with TFE3-KO mice and dual macrophage-specific TFEB-KO/TFE3-KO mice (of note: whole body TFE3-KO mice are viable). As we mentioned in our initial "response to reviewers" document, we believe this is a long-term proposition which would require us to develop these mice, phenotype them in vivo and characterize their macrophages in vitro, and then investigate the dependence of trehalose on TFEB and/or TFE3 both in vitro and in vivo. We included this limitation in the discussion section and hope the reviewer agrees with us on this point.

Our primary goal in the current manuscript was to demonstrate that induction of macrophage autophagy-lysosomal biogenesis via genetic means (TFEB overexpression) and pharmacological means (trehalose) is a viable strategy to ameliorate atherosclerosis. We conducted this using 8 separate large-scale atherosclerosis experiments and associated cell-culture experiments. Along with the extensive improvements made to the manuscript based on the very helpful comments of all the reviewers, we hope the reviewer agrees that this goal was achieved.

2. It might be worth discussing the mechanisms of reduced macrophage content in the TFEB-Tg mice lesion, e.g. through reduced monocytosis, reduced monocyte infiltration, increased clearance (efferocytosis) etc.

We have now included a paragraph in the discussion section (new 4th paragraph) regarding these possibilities.

3. The schematic figure may also incorporate that TFEB-Tg reduced lesion macrophage content, increased macrophage cholesterol efflux (partially dependent on autophagy based on Emanuel et al. ATVB paper), suppressed IL-1b secretion (independent of autophagy), increased

lysosomal acid lipase activity, but did not affect lipid uptake and accumulation based on the new data provided.

As suggested we went ahead and added these components to the schematic Figure 11 (see below on the revised schematic). We point out that this already busy schematic becomes more complicated and likely more difficult for readers to follow. Thus, we would like to keep the original figure which includes the salient findings of the manuscript and propose that a broader overview of macrophage autophagy-lysosome system in atherosclerosis is best reserved for a future review of the subject. If the reviewer feels strongly that the lipid metabolism angle should be included, we will gladly swap in the below schematic.

Reviewers #2:

The authors did an excellent job addressing this reviewers concerns. Addition of new data strengthens the conclusions. Overall, this is an important study providing significant findings that will be helpful for many researchers in the field.
No more comments.

We appreciate the assessment and thank Reviewer #2 for the initial comments that strengthened our work.

Reviewer #3:

The authors have done a very complete and remarkable job addressing all of my concerns in this manuscript. I believe that they have significantly strengthened an already important paper with their new studies.

Likewise, we appreciate the assessment and thank Reviewer #3 for the initial comments that strengthened our work.

Figure 11